# Associations of childhood experiences and methamphetamine use among Akha and Lahu hill tribe youths in northern Thailand: A cross-sectional study

Tawatchai Apidechkul [1,2]*, Chalitar Chomchoei[3], Pilasinee Wongnuch[2], Ratipark Tamornpark[1], Panupong Upala[1], Fartima Yeemard[1], Marisa Poomiphak Na Nongkhai[2], Woottichai Nachaiwieng[2], Rachanee Sunsern[2]

1 Center of Excellence for The Hill Tribe Health Research, Mae Fah Luang University, Chiang Rai, Thailand, 2 School of Health Science, Mae Fah Luang University, Chiang Rai, Thailand, 3 Chulabhorn Royal Academy, Bangkok, Thailand

* Tawatchai.api@mfu.ac.th

**Data Availability Statement:** Data is fully available in supporting information file.

## Abstract

### Background

Methamphetamine (MA) is a commonly used substance among youths, particularly those who are living in poor economic conditions with low levels of education and who have had bad childhood experiences. The Akha and Lahu hill tribe youths living on the Thailand-Myanmar-Laos border are identified as the group most vulnerable to MA use in Thailand. The study aimed to estimate the prevalence of MA use and determine its associations with childhood experiences among Akha and Lahu youths aged 15–24 years in northern Thailand.

### Methods

A cross-sectional study was performed. Validated and sealed questionnaires were used to gather information from participants after obtaining the informed consent form. Questionnaires were completed by participants and their parents at home. Logistic regression was used to identify the associations between variables at the α = 0.05 level.

### Results

A total of 710 participants participated in the study: 54.2% were Akha, 52.5% were females, 50.6% were aged 15–17 years, and 11.4% did not have Thai identification card (ID) cards. The overall prevalence of MA use at least once among Akha and Lahu youths was 14.5%. After controlling for all potential confounding factors, 8 variables were found to be associated with MA use. Males had a greater chance of MA use than females (AOR = 4.75; 95% CI = 2.27–9.95). Participants aged 21–24 years had a greater chance of MA use than those aged 15–17 years (AOR = 2.51; 95% CI = 1.11–5.71). Those who had a family member who used MA had a greater chance of MA use than those who did not (AOR = 5.04; 95% CI =

**Funding:** Funding was provided by the National Research Council of Thailand (NRCT) (Grant No.29/2561). NRCT has no role and involving any step of conducting this research.

**Competing interests:** The authors have declared that no competing interests exist.

1.66–15.32). Those who had been physically assaulted by a family member while aged 0–5 years had a greater chance of MA use than those who had not (AOR = 2.29; 95% CI = 1.02–5.12). Those who had been physically assaulted by a family member while aged 6–14 years had a greater chance of MA use than those who had not (AOR = 3.15; 95% CI = 1.32–7.54). Those who had a close friend who used alcohol had a greater chance of MA use than those who did not (AOR = 2.24; 95% CI = 1.24–4.72). Those who had a highly confident personality had a greater chance of MA use than those who did not (AOR = 2.35; 95% CI = 1.17–4.69), and those who smoked had a greater chance of MA use than those who did not (AOR = 8.27; 95% CI = 4.42–15.46).

## Conclusions

All relevant government and nongovernment agencies together with the Ministry of Public Health Thailand should address MA use among Akha and Lahu youths by properly developing a community health intervention that lowers risk of MA use by addressing family relationships, male youth behaviors, and focused on those individuals with a highly confident personality.

## Introduction

Methamphetamine (MA) has been widely recognized as the original factor contributing to several problems from individual health problems to social problems, including physical and mental health problems [1], poor family relationships [2], social problems [3], and interference with country economic growth [4]. Today, MA use is resulting in large social and economic problems globally [5]. Many communities have faced a severe stage of problems associated with MA use, particularly disruption of community economic growth [6]. The United Nations Office on Drugs and Crime (UNODC) reported that 5.6% (275 million people) of the global population aged 15–64 years used drugs at least once in 2016 [7]. In 2018, a study reported that MA was the second most commonly used illicit drug worldwide, and the Southeast Asia region was the most impacted region, including Thailand [8]. In 2019, the Department of Mental Health Ministry of Public Health, Thailand reported that there were 2.7 million Thai youths using MA [9]. Northern Thailand is considered a region of MA production and distribution due to sharing borders with Myanmar and Laos, which are recognized as the largest regions of MA production in the world [10]. Today, MA is becoming easily available in these areas due to the decrease in its price, particularly in border areas [11].

The most vulnerable group for MA use is the youth group [12]. Youths range in age from 15–24 years according to the definition of the United Nations [13]. Youths with poor socioeconomic status are reported as the group most vulnerable to MA use in Thailand, and this is particularly true among hill tribe youths [14]. The hill tribe is a group of people who have migrated into Thailand from southern China over several centuries [15] which is consisted of six main tribes: Akha, Lahu, Yao, Karen, Hmong, and Lisu [16]. The United Nations reported that most of the hill tribe people in Thailand lived below the poverty level in Thailand [17]. Akha is the largest group, followed by Lahu [16]. These two groups accounted for more than 70.0% [18] of the total hill tribe populations in Thailand, which was 3.5–4 million in 2018 [19]. Akha and Lahu have their own culture, language, and lifestyle pattern, including attitudes and perceptions toward drugs. Most Akha and Lahu villages are settled along the hill and border

areas of Thailand-Myanmar and Thailand-Laos; therefore, it is not difficult for villagers to access MA.

Childhood experiences are widely studied in various populations in different aspects, including health problems [20]. In 2014, a study in China reported that some childhood experiences were associated with MA use in adulthood [21]. A study in the United States also reported that childhood experience and household dysfunction were associated with many health problems at later ages, including death in adults [22]. People begin using MA for different purposes, such as to have fun, to get more energy to work or to be accepted by peers [23]. Akha and Lahu youths begin using MA for several reasons, such as persuasion from their peers and level of access drugs [24]. However, there is no study detecting the associations between childhood experiences and MA use, particularly in hill tribe youths in Thailand, who are recognized as one of the most vulnerable populations for MA use.

Therefore, the study aimed to estimate the prevalence of MA use and determine the association of childhood experience with MA use among Akha and Lahu youths aged 15–24 years who lived in northern Thailand. The findings could be used to develop public health interventions for reducing MA use among Akha and Lahu hill tribe youths in Thailand.

## Methods

### Study design

A cross-sectional study design was applied to collect data from the selected participants.

### Study population

The study population included Akha and Lahu youths aged 15–24 years who lived in Chiang Rai Province, Thailand, in 2019.

### Study sample

The study sample was composed of Akha and Lahu youths aged 15–24 years who lived in Chiang Rai Province in 2019 and were randomly selected for the study. However, those who could not identify themselves as members of the Akha or Lahu tribes were excluded from the study. Moreover, both participants and parents who could not provide essential information regarding the study protocols were also excluded from the study.

### Sample size

The sample size was calculated by the standard formula for a cross-sectional study design [25]. After the calculation, based on the assumption of p = 0.27 [26], q = 0.73, and e = 0.05, there were 670 participants required for the analysis: approximately 335 for the Lahu tribe and another 335 for the Akha tribe.

Since, there is no scientific data available on the prevalence of the MA use among the hill tribe population, then, the calculation for the sample size was based on the information (prevalence) from the study conducted in Thai youth who lived in the central of Bangkok which was conducted by Toeam, et al [26]. Moreover, based on the information of the number of populations between the Akha and Lahu which was reported by the Hill tribe Welfare and Development Center [18], two tribes had similar size of the population living in 243 Akah villages (approximately 60,000 population) and 216 Lahu villages (approximately 50,000 population).

## Steps of data collection

In 2018, there were 243 Akha villages and 216 Lahu villages in Chiang Rai Province. Ten villages from each tribe were randomly selected by a simple random method. Government officers who were responding to the selected villages were asked for approval to perform the study in the targeted villages. After obtaining approval for access to the villages from the district officers, cooperation from the village headmen was obtained before collecting data. The lists of youths aged 15–24 years in the selected villages were obtained from the village headmen. All eligible individuals according to the lists received from the village headman were invited to participate in the study: 496 people from 10 Akha villages and 518 people from 10 Lahu villages. Appointments were made five days before assessing the participants to complete the questionnaire.

All participants were provided all information regarding the study protocols, particularly the security of the information obtained from all participants. Informed consent was obtained before completion of the questionnaire. Questionnaires were packed and sealed before being provided to the participants. Questionnaires were completed by participants in a personal private place and returned to researchers the next day. All questions in part one, two, four, five, and six were completed by participants (children) including part of socio-economic status of the family (part two), and nobody knew the content before reaching to researcher. However, questions in part three were separated and completed by their parents. A few people could not completely use Thai, then they were interviewed by researcher to complete the questionnaire. The reason to collect the information on experience of violence during individual's aged 0–5 years, from parents was to improve the quality of the information. This protocol was proved from the pilot phase. The process of completion of the questionnaire was blinded, and no information could be referred back to any individual. The questionnaires were sent back to the researcher on the next day and completely sealed. All questionnaires were destroyed properly after coding. Data file was kept with security code.

## Research instruments

The questionnaire was developed based on the literature and discussion with five experts who were working in the fields of youth and child behaviors (3 people) and behaviors related to MA use (2 people) including the findings from our previous study [24]. The questionnaire consisted of six parts. In part one, 10 questions were used to collect data on general information such as age, sex, tribe, marital status, etc. In part two, 16 questions were used to collect information on the family, such as the relationship of the parents, number of family members, monthly family income, etc. In part three, 11 questions were used to collect an individual's experience from 0–5 years of age, including abuse experience, such as history of assault and abuse from family members, abuse from peers in school, and sexual abuse. In this section, all information was obtained from the parents. In part four, 13 questions were used to collect information on history of being assaulted or abused while aged 6–14 years, including a history of school expulsion, assault by family members, assault by their peers, etc. In part five, 26 questions were used to collect information on personal behaviors such as smoking behavior, alcohol use, amphetamine use, etc. In part six, 20 questions were used to collect information on knowledge and attitude toward MA use. At the end of questionnaire, it appeared a short question on asking the experience in use of MA.

Subsequently, the questionnaire was examined for content validity by the item-objective congruence (IOC) technique, which was executed by three external experts in relevant fields: public health, psychology, and psychiatry. The feasibility and reliability of the questionnaire were detected by piloting with 10 selected Akha youths (5 males and 5 females) and another 10

selected Lahu youths (5 males and 5 females). The questionnaires were conducted three (3) times in the same piloting samples before being ready for use in the field. The sequencing and appropriateness of the questions were tested in the first and second rounds of the pilot. The last round was aimed at testing the reliability, which was found to have a Cronbach's alpha of 0.78. The process of filling the questionnaire lasted 25 minutes for youth and 10 minutes for parents.

## Statistical analysis

Data were coded and double entered into an Excel file. Data files were transferred into SPSS version 24 (SPSS, Chicago, IL) for analysis. Descriptive data analysis was performed; categorical data were described in percentages. The means and its standard deviations (SDs) were used to describe the characteristics of continuous data. Logistic regression was used to detect the associations of childhood experiences with MA use among the Akha and Lahu youths at the significance level of alpha 0.05. The "Enter" mode was used in the step of selection independent variables into the statistical model. The pseudo $R^2$ of Cox-Snell $R^2$ and Nagelkerke $R^2$, and the Hosmer- Lemshow chi-square were used to determine the fit of the model in all steps. Some variables were controlled the effect in the model which were determined as the confounder factors for the prediction. In the final model, all significant variables and controlled variables were fitted before making interpretations.

## Ethical approval and consent to participate

All research concept, procedures, and instruments were approved by the Mae Fah Luang University Research Ethic Committee on Human Research (REH-60141). Participants were asked their wiliness to participate the study by obtaining written informed consent form before completion the questionnaire in a private and confident room. Among those participants aged less than 18 years, parents were asked to agree in providing information in the questionnaire on behalf of their children by signing on the informed consent.

## Results

The participation rate was 77.6% (385 out of 496) in Akha, and 62.7% (325 out of 518) in Lahu. A total of 710 participants participated in the study; 54.2% were Akha, 52.5% were females, 50.6% were aged 15–17 years (mean = 18.1, SD = 2.7), and 11.4% did not have Thai ID cards. The majority were single (91.8%) and Christian (60.1%). Most participants had a high school and lower education (84.8%), lived with their parents (63.7%), and had 4–6 family members (66.9%) (Table 1).

   More than half of the participants had a family member who smoked (52.4%) and used alcohol (56.3%), while a few participants had a family member who used other substances. In the comparison analysis in experiences of family members on exposing to drugs and alcohol use between two tribes, it was found that no variable was found statistical significance (Table 2).

   The majority of caregivers while the participants were aged 0–5 years were mothers (72.4%), and most of the participants were supported by their family (58.2%). A few people had accidents (16.8%) and were hospitalized (29.4%) due to a health problem. Eighty-seven participants (12.3%) were assaulted by family members, and 15.6% were assaulted by peers in school. While having a comparison between tribes in the potential exposures relevant to MA use while aged 0–5 years, four variable were found the statistical differences; main care giver (p-value = 0.014), having accident (p-value = 0.035), having been hospitalized (p-

**Table 1. General characteristics of the participants.**

| Characteristics | Total n (%) | Akha n (%) | Lahu n (%) |
|---|---|---|---|
| **Total** | 710 (100.0) | 385 (54.2) | 325 (45.8) |
| **Sex** | | | |
| Male | 337 (47.5) | 190 (49.4) | 147 (45.2) |
| Female | 373 (52.5) | 195 (50.6) | 178 (54.8) |
| **Age** (years) | | | |
| 15–17 | 359(50.5) | 208 (54.0) | 151 (46.5) |
| 18–20 | 216 (30.4) | 119 (30.9) | 97 (29.8) |
| 21–24 | 135 (19.0) | 58 (15.1) | 77 (23.7) |
| Mean = 18.04, SD = 2.67 | | | |
| **Marital status** | | | |
| Single | 652(91.8) | 372 (96.6) | 280 (86.2) |
| Married | 55(7.5) | 12 (3.1) | 43 (13.2) |
| Other | 3(0.3) | 1 (0.3) | 2 (0.6) |
| **Religion** | | | |
| Buddhist | 283(39.9) | 128 (33.2) | 155 (47.7) |
| Christian | 427(60.1) | 257 (66.8) | 170 (52.3) |
| **Education** | | | |
| No educated | 71(10.0) | 22 (5.7) | 49 (15.1) |
| Primary school | 76(10.7) | 18 (4.7) | 58 (17.8) |
| Secondary school | 163(23.0) | 87 (22.6) | 76 (23.4) |
| High school | 292(41.1) | 209 (54.3) | 83 (25.5) |
| Vocational and university | 108(15.2) | 49 (12.8) | 59 (18.2) |
| **Occupation** | | | |
| Student | 448(63.1) | 288 (74.8) | 160 (49.2) |
| Employed | 126(17.7) | 50 (12.9) | 76 (23.4) |
| Agriculturist | 17(2.4) | 8 (2.1) | 9 (2.8) |
| Unemployed | 119(16.8) | 39 (10.1) | 80 (24.6) |
| **Thai identification** (ID) **card** | | | |
| Yes | 629(88.6) | 339 (88.1) | 290 (89.2) |
| No | 81(11.4) | 46 (11.9) | 35 (10.8) |
| **Village location** | | | |
| Rural | 352(49.6) | 176 (45.7) | 176 (54.2) |
| Semi-urban | 358(50.4) | 209 (54.3) | 149 (45.8) |
| **Living with** | | | |
| Parents | 452(63.7) | 241 (62.6) | 211 (64.9) |
| Father | 49(6.9) | 29 (7.5) | 20 (6.2) |
| Mother | 86(12.1) | 59 (15.3) | 87 (8.3) |
| Stepfather or stepmother | 19(2.7) | 16 (4.2) | 3 (0.9) |
| Relatives | 104(14.6) | 40 (10.4) | 64 (19.7) |
| **Parents' status** | | | |
| Married and living together | 510(71.8) | 272 (70.6) | 238 (73.2) |
| Either father or mother died | 61(8.6) | 38 (9.9) | 23 (7.1) |
| Both father and mother died | 13(1.8) | 6 (1.5) | 7 (2.2) |
| Separated | 60(8.5) | 38 (9.9) | 22 (6.7) |
| Divorced | 66(9.3) | 31 (8.1) | 35 (10.8) |
| **Number of family members** (people) | | | |
| ≤ 3 | 119(16.8) | 65 (16.9) | 54 (16.6) |

(*Continued*)

**Table 1.** (Continued)

| Characteristics | Total n (%) | Akha n (%) | Lahu n (%) |
|---|---|---|---|
| 4–6 | 475(66.9) | 246 (63.9) | 229 (70.5) |
| ≥ 7 | 116(16.3) | 74 (19.2) | 42 (12.9) |
| **Family income per month** (baht) | | | |
| ≤ 10,000 | 213(30.0) | 104 (27.0) | 109 (33.5) |
| 10,001–20,000 | 44(6.2) | 26 (6.8) | 18 (5.5) |
| ≥ 20,001 | 44(6.2) | 32 (8.3) | 12 (3.7) |
| Unknown | 409(57.6) | 223 (57.9) | 186 (57.3) |

value = 0.015), and having been physically assaulted by peer in school (p-value = 0.025) (Table 3).

While participants were in the age range of 6–14 years, 65.4% were cared for by their mother, and few people had been supported by their family in regard to getting desirable food (5.2%) and travelling to desirable places (8.2%). Almost one-third had a head injury (16.2%). A few people were assaulted by family members (8.5%), assaulted due to their sexual orientation (4.1%), assaulted due to their socioeconomic status by their peers in school (12.1%), and sexually abused (1.3%). In the comparison analysis between tribes in the aspect of having the potential exposures relevant to MA use while aged 6–14 years, three (3) variable were found the statistical differences; having had travelled to desired places with support from parents (p-value = 0.02), had been greatly supported by parents in regards to desired clothes and other items (p-value = 0.048), and failed class examination (p-value = 0.010) (Table 4).

One hundred and three participants (14.5%) reported that they had used MA at least once in their life, 18.5% smoked, and 36.1% used alcohol. Most participants used Facebook (93.5%)

**Table 2. Experiences of family members on exposing to drugs and alcohol.**

| Exposure | Total | | Akha | | Lah | | $\chi^2$ (p-value) |
|---|---|---|---|---|---|---|---|
| | n | % | n | % | n | % | |
| **Having family member who ever smoked** | | | | | | | |
| No | 338 | 47.6 | 191 | 49.6 | 147 | 45.2 | 1.36 (0.244) |
| Yes | 372 | 52.4 | 194 | 50.4 | 178 | 54.8 | |
| **Having family member who ever used alcohol** | | | | | | | |
| No | 310 | 43.7 | 158 | 41.0 | 152 | 46.8 | 2.35 (0.125) |
| Yes | 400 | 56.3 | 227 | 59.0 | 173 | 53.2 | |
| **Having family member who ever used glue** | | | | | | | |
| No | 690 | 97.2 | 378 | 98.2 | 312 | 96.0 | 3.07 (0.080) |
| Yes | 20 | 2.8 | 7 | 1.8 | 13 | 4.0 | |
| **Having family member who ever used methamphetamine** | | | | | | | |
| No | 684 | 96.3 | 371 | 96.4 | 313 | 96.3 | 0.02 (0.968) |
| Yes | 26 | 3.7 | 14 | 3.6 | 12 | 3.7 | |
| **Having family member who ever used heroin** | | | | | | | |
| No | 696 | 98.0 | 377 | 97.9 | 319 | 98.2 | 0.05 (0.825) |
| Yes | 14 | 2.0 | 8 | 2.1 | 6 | 1.8 | |
| **Having family member who ever used opium** | | | | | | | |
| No | 687 | 96.8 | 376 | 97.7 | 311 | 95.7 | 2.18 (0.140) |
| Yes | 23 | 3.2 | 9 | 2.3 | 14 | 4.3 | |

*Significant level at α = 0.05

**Table 3. Potential exposures relevant to MA use while aged 0–5 years.**

| Family information | Total | | Akha | | Lahu | | $\chi^2$ (p-value) |
|---|---|---|---|---|---|---|---|
| | n | % | n | % | n | % | |
| **Main caregiver** | | | | | | | |
| Mother | 514 | 72.4 | 268 | 69.6 | 246 | 75.7 | 14.34 (0.014*) |
| Father | 87 | 12.3 | 62 | 16.1 | 25 | 7.7 | |
| Stepfather | 10 | 1.4 | 3 | 0.8 | 7 | 2.2 | |
| Stepmother | 11 | 1.5 | 7 | 1.8 | 4 | 1.2 | |
| Other relative | 88 | 12.4 | 45 | 11.7 | 43 | 13.2 | |
| **Used to be greatly supported by parents in regard to receiving desired food and beverages** | | | | | | | |
| No | 644 | 90.7 | 354 | 91.9 | 290 | 89.2 | 1.54 (0.214) |
| Yes | 66 | 9.3 | 31 | 8.1 | 35 | 10.8 | |
| **Used to travel to desired places with support from parents** | | | | | | | |
| No | 556 | 78.3 | 306 | 79.5 | 250 | 76.9 | 0.68 (0.410) |
| Yes | 154 | 21.7 | 79 | 20.5 | 75 | 23.1 | |
| **Used to be greatly supported by parents in regard to desired clothes and other items** | | | | | | | |
| No | 610 | 85.9 | 339 | 88.1 | 271 | 83.4 | 3.17 (0.075) |
| Yes | 100 | 14.1 | 46 | 11.9 | 54 | 16.6 | |
| **Accident** | | | | | | | |
| No | 591 | 83.2 | 310 | 80.5 | 281 | 86.5 | 4.46 (0.035*) |
| Yes | 119 | 16.8 | 75 | 19.5 | 44 | 13.5 | |
| **Hospitalization** | | | | | | | |
| No | 501 | 70.6 | 257 | 66.8 | 244 | 75.1 | 5.88 (0.015*) |
| Yes | 209 | 29.4 | 128 | 33.2 | 81 | 24.9 | |
| **Head injury** | | | | | | | |
| No | 600 | 84.5 | 326 | 84.7 | 274 | 84.3 | 0.02 (0.893) |
| Yes | 110 | 15.5 | 59 | 15.3 | 51 | 15.7 | |
| **Physically assaulted by family member** | | | | | | | |
| No | 623 | 87.7 | 337 | 87.5 | 286 | 88.0 | 0.04 (0.850) |
| Yes | 87 | 12.3 | 48 | 12.5 | 39 | 12.0 | |
| **Physically assaulted by peer in school** | | | | | | | |
| No | 599 | 84.4 | 314 | 81.6 | 285 | 87.7 | 5.03 (0.025*) |
| Yes | 111 | 15.6 | 71 | 18.4 | 40 | 12.3 | |

*Significant level at $\alpha$ = 0.05

and the Line application (73.8%). More than one-fourth (25.9%) had their urine tested for MA by police officer, and 7.6% had been arrested. Most participants had ≤5 close friends (78.6%), and of those close friends; 18.0% smoked, 27.5% used alcohol, and 2.1% used MA. The majority had an active and talkative personality (63.2%), were social (81.3%), and had high self-confidence (70.4%). While in the comparisons between tribes in behaviors and personalities, twelve (12) variables were found the statistical differences; regularly exercise (p-value = 0.001), ever played online games (p-value = 0.001), ever used Facebook (p-value = 0.001), frequency of Facebook use (p-value = 0.001), ever used the Line application (p-value = 0.004), ever tested for MA in urine by police officer (p-value<0.001), number of close friends (p-value<0.001), close friend who drink alcohol (p-value = 0.039), close friend who uses MA (p-value = 0.030), personality (p-value = 0.017), highly self-confident behavior (p-value = 0.014), and socialized behavior (p-value = 0.032) (Table 5).

**Table 4. Potential exposures relevant to MA use while aged 6–14 years.**

| Exposure | Total | | Akha | | Lahu | | $\chi^2$ (p-value) |
|---|---|---|---|---|---|---|---|
| | n | % | n | % | n | % | |
| **Main caregiver** | | | | | | | |
| Mother | 464 | 65.4 | 250 | 64.9 | 214 | 65.8 | 6.25 (0.283) |
| Father | 113 | 15.9 | 70 | 18.2 | 43 | 13.2 | |
| Stepfather | 7 | 1.0 | 2 | 0.5 | 5 | 1.5 | |
| Stepmother | 15 | 2.1 | 8 | 2.1 | 7 | 2.2 | |
| Relatives | 111 | 15.6 | 55 | 14.3 | 56 | 17.3 | |
| **Used to be greatly supported by parents in regard to receiving desired food and beverages** | | | | | | | |
| No | 673 | 94.8 | 367 | 95.3 | 306 | 94.2 | 0.49 (0.484) |
| Yes | 37 | 5.2 | 18 | 4.7 | 19 | 5.8 | |
| **Travelled to desired places with support from parents** | | | | | | | |
| No | 652 | 91.8 | 365 | 94.8 | 287 | 88.3 | 9.92 (0.002*) |
| Yes | 58 | 8.2 | 20 | 5.2 | 38 | 11.7 | |
| **Had been greatly supported by parents in regard to desired clothes and other items** | | | | | | | |
| No | 45 | 6.3 | 367 | 95.3 | 298 | 91.7 | 3.92 (0.048*) |
| Yes | 665 | 93.7 | 18 | 4.9 | 27 | 8.3 | |
| **Accident** | | | | | | | |
| No | 578 | 81.4 | 311 | 80.8 | 267 | 82.2 | 0.22 (0.639) |
| Yes | 132 | 18.6 | 74 | 19.2 | 58 | 17.8 | |
| **Hospitalization** | | | | | | | |
| No | 557 | 78.5 | 293 | 76.1 | 264 | 81.2 | 2.74 (0.098) |
| Yes | 153 | 21.5 | 92 | 23.9 | 61 | 18.8 | |
| **Head injury** | | | | | | | |
| No | 595 | 83.8 | 315 | 81.8 | 280 | 86.2 | 2.44 (0.118) |
| Yes | 115 | 16.2 | 70 | 18.2 | 45 | 13.8 | |
| **Expulsion from school** | | | | | | | |
| No | 693 | 97.6 | 374 | 97.1 | 319 | 98.2 | 0.77 (0.380) |
| Yes | 17 | 2.4 | 11 | 2.9 | 6 | 1.8 | |
| **Assaulted by family member** | | | | | | | |
| No | 650 | 91.5 | 354 | 91.9 | 296 | 91.1 | 0.17 (0.678) |
| Yes | 60 | 8.5 | 31 | 8.1 | 29 | 8.9 | |
| **Assaulted by peer in school** | | | | | | | |
| No | 622 | 87.6 | 330 | 85.7 | 292 | 89.8 | 2.77 (0.096) |
| Yes | 88 | 12.4 | 55 | 14.3 | 33 | 10.2 | |
| **Insulted due to sexual orientation** | | | | | | | |
| No | 681 | 95.9 | 372 | 96.6 | 309 | 95.1 | 1.08 (0.300) |
| Yes | 29 | 4.1 | 13 | 3.4 | 16 | 4.9 | |
| **Insulted due to socioeconomic status** | | | | | | | |
| No | 624 | 87.9 | 339 | 88.1 | 285 | 87.7 | 0.02 (0.884) |
| Yes | 86 | 12.1 | 46 | 11.9 | 40 | 12.3 | |
| **Sexually abused** | | | | | | | |
| No | 697 | 98.2 | 379 | 98.4 | 318 | 97.8 | 0.35 (0.555) |
| Yes | 13 | 1.8 | 6 | 1.6 | 7 | 2.2 | |
| **Failed class examination** | | | | | | | |
| No | 420 | 59.2 | 211 | 54.8 | 209 | 64.3 | 6.59 (0.010*) |
| Yes | 290 | 40.8 | 174 | 45.2 | 116 | 35.7 | |

*Significant level at α = 0.05

**Table 5. Participants' behaviors and personality.**

| Characteristics | Total | | Akha | | Lahu | | χ²(p-value) |
|---|---|---|---|---|---|---|---|
| | n | % | n | % | n | % | |
| **Ever used MA at least once** | | | | | | | |
| No | 607 | 85.5 | 330 | 85.7 | 277 | 85.2 | 0.03 (0.855) |
| Yes | 103 | 14.5 | 55 | 14.3 | 48 | 14.8 | |
| **Ever smoked** | | | | | | | |
| No | 579 | 81.5 | 307 | 79.7 | 272 | 83.7 | 1.83 (0.176) |
| Yes | 131 | 18.5 | 78 | 20.3 | 53 | 16.3 | |
| **Ever used alcohol** | | | | | | | |
| No | 454 | 63.9 | 235 | 61.0 | 219 | 67.4 | 3.08 (0.079) |
| Yes | 256 | 36.1 | 150 | 39.0 | 106 | 32.6 | |
| **Regularly exercise** | | | | | | | |
| No | 90 | 12.7 | 29 | 7.5 | 61 | 18.8 | 20.10 (0.001*) |
| Yes | 620 | 87.3 | 356 | 92.5 | 264 | 81.2 | |
| **Ever played online games** | | | | | | | |
| No | 298 | 42.0 | 138 | 35.8 | 160 | 49.2 | 12.97 (0.001*) |
| Yes | 412 | 58.0 | 247 | 64.8 | 165 | 50.8 | |
| **Ever used Facebook** | | | | | | | |
| No | 46 | 6.5 | 13 | 3.4 | 33 | 10.2 | 13.36 (0.001*) |
| Yes | 664 | 93.5 | 372 | 96.6 | 292 | 89.8 | |
| **Frequency of Facebook use** (n = 664) | | | | | | | |
| Sometimes | 145 | 21.8 | 58 | 15.6 | 87 | 29.8 | 24.73 (0.001*) |
| Often | 203 | 30.6 | 110 | 29.6 | 93 | 31.8 | |
| Everyday | 316 | 47.6 | 204 | 54.8 | 112 | 38.4 | |
| **Ever used the Line Application** | | | | | | | |
| No | 186 | 26.2 | 84 | 21.8 | 102 | 31.4 | 8.34 (0.004*) |
| Yes | 524 | 73.8 | 301 | 78.2 | 223 | 68.6 | |
| **Frequency of use of the Line Application** (n = 524) | | | | | | | |
| Sometimes | 202 | 38.5 | 117 | 38.9 | 85 | 38.1 | 1.87 (0.392) |
| Often | 175 | 33.4 | 94 | 31.2 | 81 | 36.3 | |
| Everyday | 147 | 28.1 | 90 | 29.9 | 57 | 25.6 | |
| **Experienced a broken heart** | | | | | | | |
| No | 339 | 47.7 | 181 | 47.0 | 158 | 48.6 | 0.18 (0.670) |
| Yes | 371 | 52.3 | 204 | 53.0 | 167 | 51.4 | |
| **Used to work in the night-work sector** | | | | | | | |
| No | 688 | 96.9 | 375 | 97.4 | 313 | 96.3 | 0.70 (0.402) |
| Yes | 22 | 3.1 | 10 | 2.6 | 12 | 3.7 | |
| **Used to have sex in exchange for items or money** | | | | | | | |
| No | 700 | 98.6 | 377 | 97.9 | 323 | 99.4 | 2.72 (0.099) |
| Yes | 10 | 1.4 | 8 | 2.1 | 2 | 0.6 | |
| **Ever tested for MA in urine by police officer** | | | | | | | |
| No | 526 | 74.1 | 257 | 66.8 | 269 | 82.8 | 23.55 (<0.001*) |
| Yes | 184 | 25.9 | 128 | 33.2 | 56 | 17.2 | |
| **Arrested** | | | | | | | |
| No | 656 | 92.4 | 357 | 92.7 | 299 | 92.0 | 0.13 (0.716) |
| Yes | 54 | 7.6 | 28 | 7.3 | 26 | 8.0 | |
| **Number of close friends** (people) | | | | | | | |

(*Continued*)

**Table 5.** (Continued)

| Characteristics | Total | | Akha | | Lahu | | $\chi^2$ (p-value) |
|---|---|---|---|---|---|---|---|
| | n | % | n | % | n | % | |
| $\leq 5$ | 558 | 78.6 | 280 | 72.7 | 278 | 85.5 | 17.33 (<0.001*) |
| 6–10 | 141 | 19.9 | 98 | 25.5 | 43 | 13.2 | |
| $\geq 11$ | 11 | 1.5 | 7 | 1.8 | 4 | 1.2 | |
| **Close friend who smokes** | | | | | | | |
| No | 582 | 82.0 | 306 | 79.5 | 276 | 84.9 | 3.53 (0.060) |
| Yes | 128 | 18.0 | 79 | 20.5 | 49 | 15.1 | |
| **Close friend who drinks alcohol** | | | | | | | |
| No | 515 | 72.5 | 267 | 69.4 | 248 | 76.3 | 4.28 (0.039*) |
| Yes | 195 | 27.5 | 118 | 30.6 | 77 | 23.7 | |
| **Close friend who uses glue** | | | | | | | |
| No | 695 | 97.9 | 379 | 98.4 | 316 | 97.2 | 1.25 (0.264) |
| Yes | 15 | 2.1 | 6 | 1.6 | 9 | 2.8 | |
| **Close friend who uses heroin** | | | | | | | |
| No | 701 | 98.7 | 382 | 99.2 | 319 | 98.2 | 1.60 (0.205) |
| Yes | 9 | 1.3 | 3 | 0.8 | 6 | 1.8 | |
| **Close friend who uses MA** | | | | | | | |
| No | 695 | 97.9 | 381 | 99.0 | 314 | 96.6 | 4.69 (0.030*) |
| Yes | 15 | 2.1 | 4 | 1.0 | 11 | 3.4 | |
| **Personality** | | | | | | | |
| Polite and quiet | 200 | 28.2 | 100 | 26.0 | 100 | 30.8 | 8.20 (0.017*) |
| Active and talkative | 449 | 63.2 | 260 | 67.5 | 189 | 58.2 | |
| Stay alone | 61 | 8.6 | 25 | 6.5 | 36 | 11.1 | |
| **Highly self-confident behavior** | | | | | | | |
| No | 210 | 29.6 | 99 | 25.7 | 111 | 34.2 | 6.03 (0.014*) |
| Yes | 500 | 70.4 | 286 | 74.3 | 214 | 65.8 | |
| **Socialized behavior** | | | | | | | |
| No | 133 | 18.7 | 61 | 15.8 | 72 | 22.2 | 4.61 (0.032*) |
| Yes | 577 | 81.3 | 324 | 84.2 | 253 | 77.8 | |

* Significant level at $\alpha = 0.05$

In the univariate analysis that was performed to identify factors associated with MA use among the Akha and Lahu hill tribe youths, there were several factors associated with MA use, such as sex, age, occupation, parents' marital status, number of family members, family member smoking status, family member alcohol use, and family member amphetamine use (Table 6).

After controlling for tribe, marital status, religion, education, occupation, and having Thai ID card in the multivariate model, 8 variables were found to be associated with MA use among the Akha and Lahu youths in northern Thailand: sex, age, having a family member who used MA, having been physical assaulted by family member while aged 0–5 years, having been physical assaulted by family member while aged 6–14 years, having a close friend who drinks alcohol, having a highly confident personality, smoking.

Males had a 4.75-fold (95% CI = 2.27–9.95) greater chance of MA use than females. Participants aged 21–24 years had a 2.51-fold (95% CI = 1.11–5.71) greater chance of MA use than those aged 15–17 years. Those who had a family member who used MA had a 5.04-fold (95% CI = 1.66–15.32) greater chance of MA use than those who did not. Those who had been

**Table 6. Univariate and multivariate analyses of factors associated with MA use among Akha and Lahu youths.**

| Factor | MA use | | | | Univariate analysis | | | Multivariate analysis | | |
|---|---|---|---|---|---|---|---|---|---|---|
| | Yes | | No | | OR | 95% CI | p-value | AOR | 95% CI | p-value |
| | n | % | n | % | | | | | | |
| Total | 103 | 14.5 | 607 | 85.5 | N/A | N/A | N/A | N/A | N/A | N/A |
| **Sex** | | | | | | | | | | |
| Male | 91 | 27.0 | 246 | 73.0 | 11.13 | 5.97–20.76 | <0.001* | 4.75 | 2.27–9.95 | <0.001* |
| Female | 12 | 3.2 | 361 | 96.8 | 1.00 | | | 1.00 | | |
| **Age** (years) | | | | | | | | | | |
| 15–17 | 35 | 9.7 | 324 | 90.3 | 1.00 | | | 1.00 | | |
| 18–20 | 35 | 16.2 | 181 | 83.8 | 1.79 | 1.08–2.96 | 0.023* | 1.90 | 0.98–3.69 | 0.059 |
| 21–24 | 33 | 24.4 | 102 | 75.6 | 3.00 | 1.77–5.06 | <0.001* | 2.51 | 1.11–5.71 | 0.028* |
| **Tribe** | | | | | | | | | | |
| Akha | 55 | 14.3 | 330 | 85.7 | | | | | | |
| Lahu | 48 | 14.8 | 277 | 85.2 | 1.04 | 0.68–1.58 | 0.855 | | | |
| **Marital status** | | | | | | | | | | |
| Single | 96 | 14.7 | 556 | 85.3 | 1.00 | | | | | |
| Married | 6 | 10.9 | 49 | 89.1 | 2.90 | 0.26–32.25 | 0.387 | | | |
| Other | 1 | 33.3 | 2 | 66.7 | 0.71 | 0.30–1.70 | 0.441 | | | |
| **Religion** | | | | | | | | | | |
| Buddhist | 44 | 15.5 | 239 | 84.5 | 1.00 | | | | | |
| Christian | 59 | 13.8 | 368 | 86.2 | 0.87 | 0.57–1.33 | 0.522 | | | |
| **Education** | | | | | | | | | | |
| Non-educated | 19 | 26.8 | 52 | 73.2 | 4.39 | 0.53–36.04 | 0.169 | | | |
| Primary school | 20 | 26.3 | 56 | 73.7 | 4.23 | 0.52–35.10 | 0.175 | | | |
| Secondary school | 21 | 12.9 | 142 | 87.1 | 1.78 | 0.22–14.36 | 0.591 | | | |
| High school | 36 | 12.3 | 256 | 87.7 | 1.69 | 0.21–13.37 | 0.620 | | | |
| Vocational and university | 7 | 6.5 | 101 | 93.5 | 1.00 | | | | | |
| **Occupation** | | | | | | | | | | |
| Student | 37 | 8.3 | 411 | 91.7 | 1.00 | | | | | |
| Employed | 31 | 24.6 | 95 | 75.4 | 3.66 | 2.13–6.31 | <0.001* | | | |
| Agriculturalist | 4 | 23.5 | 13 | 76.5 | 3.42 | 1.03–10.98 | 0.040* | | | |
| Unemployed | 31 | 26.1 | 88 | 73.9 | 3.91 | 2.30–6.65 | <0.001* | | | |
| **Thai identification card** | | | | | | | | | | |
| Yes | 97 | 15.4 | 532 | 84.6 | 1.00 | | | | | |
| No | 6 | 7.4 | 75 | 92.6 | 2.28 | 0.97–5.38 | 0.060 | | | |
| **Village location** | | | | | | | | | | |
| Rural | 45 | 12.8 | 307 | 87.2 | 1.00 | | | | | |
| Semiurban | 58 | 16.2 | 300 | 83.8 | 1.32 | 0.87–2.01 | 0.197 | | | |
| **Parents' marital status** | | | | | | | | | | |
| Living together | 57 | 11.2 | 453 | 88.8 | 1.00 | | | | | |
| Either father or mother died | 15 | 24.6 | 46 | 75.4 | 2.59 | 1.36–4.94 | 0.004* | | | |

*(Continued)*

**Table 6.** (Continued)

| Factor | MA use | | | | Univariate analysis | | | Multivariate analysis | | |
|---|---|---|---|---|---|---|---|---|---|---|
| | Yes | | No | | OR | 95% CI | p-value | AOR | 95% CI | p-value |
| | n | % | n | % | | | | | | |
| Both father and mother died | 1 | 7.7 | 12 | 92.3 | 0.66 | 0.09–5.19 | 0.695 | | | |
| Separated | 16 | 26.7 | 44 | 73.3 | 2.89 | 1.53–5.45 | 0.001* | | | |
| Divorced | 14 | 21.2 | 52 | 78.8 | 2.14 | 1.12–4.10 | 0.022* | | | |
| **Number of family members** (people) | | | | | | | | | | |
| ≤ 3 | 25 | 21.0 | 94 | 79.0 | 3.16 | 1.41–7.11 | 0.005* | | | |
| 4–6 | 69 | 14.5 | 406 | 84.5 | 2.02 | 0.98–4.18 | 0.058 | | | |
| ≥ 7 | 9 | 7.8 | 107 | 92.2 | 1.00 | | | | | |
| **Family income per month** (baht) | | | | | | | | | | |
| ≤ 10,000 | 37 | 17.4 | 176 | 82.6 | 1.00 | | | | | |
| 10,001–20,000 | 7 | 15.9 | 37 | 84.1 | 0.90 | 0.37–2.17 | 0.815 | | | |
| ≥ 20,001 | 6 | 13.6 | 38 | 86.4 | 0.75 | 0.30–1.01 | 0.547 | | | |
| Unknown | 53 | 13.0 | 356 | 87.0 | 0.71 | 0.45–1.12 | 0.139 | | | |
| **Having a family member who smokes** | | | | | | | | | | |
| No | 30 | 8.9 | 308 | 91.1 | 1.00 | 1.59–3.95 | <0.001* | | | |
| Yes | 73 | 19.6 | 299 | 80.4 | 2.51 | | | | | |
| **Having a family member who uses alcohol** | | | | | | | | | | |
| No | 34 | 11.0 | 276 | 89.0 | 1.00 | | | | | |
| Yes | 69 | 17.3 | 331 | 82.8 | 1.69 | 1.09–2.63 | 0.019* | | | |
| **Having a family member who uses glue** | | | | | | | | | | |
| No | 93 | 13.5 | 597 | 86.5 | 1.00 | | | | | |
| Yes | 10 | 50.0 | 10 | 50.0 | 6.42 | 2.60–15.84 | <0.001* | | | |
| **Having family member who uses methamphetamine** | | | | | | | | | | |
| No | 89 | 13.0 | 595 | 87.0 | 1.00 | | | 1.00 | | |
| Yes | 14 | 53.8 | 12 | 46.2 | 7.80 | 3.50–17.40 | <0.001* | 5.04 | 1.66–15.32 | 0.004 |
| **Having a family member who uses heroin** | | | | | | | | | | |
| No | 98 | 14.1 | 598 | 85.9 | 1.00 | | | | | |
| Yes | 5 | 35.7 | 9 | 64.3 | 3.39 | 1.11–10.33 | 0.032* | | | |
| **Having a family member who uses opium** | | | | | | | | | | |
| No | 95 | 13.8 | 592 | 86.2 | 1.00 | | | | | |
| Yes | 8 | 34.8 | 15 | 65.2 | 3.32 | 1.37–8.05 | 0.008* | | | |
| **Main caregiver from the ages of 0–5 years** | | | | | | | | | | |
| Mother | 63 | 12.3 | 451 | 87.7 | 1.00 | | | | | |
| Father | 17 | 19.5 | 70 | 80.5 | 1.74 | 0.96–3.14 | 0.067 | | | |
| Stepfather | 4 | 40.0 | 6 | 60.0 | 4.77 | 1.31–17.38 | 0.018 | | | |
| Stepmother | 3 | 27.3 | 8 | 72.7 | 2.69 | 0.69–10.39 | 0.153 | | | |
| Other relative | 16 | 18.2 | 72 | 81.8 | 1.59 | 0.84–3.01 | 0.153 | | | |
| **Used to be greatly supported in regard to receiving desirable food and beverage from parents while aged 0–5 years** | | | | | | | | | | |
| No | 11 | 16.7 | 55 | 83.3 | 1.00 | | | | | |
| Yes | 92 | 14.3 | 552 | 85.7 | 1.20 | 0.61–2.38 | 0.601 | | | |
| **Used to be greatly supported by parents in regard to travelling to desirable places while aged 0–5 years** | | | | | | | | | | |
| No | 27 | 17.5 | 127 | 82.5 | 1.00 | | | | | |

(Continued)

**Table 6.** (Continued)

| Factor | MA use | | | | Univariate analysis | | | Multivariate analysis | | |
|---|---|---|---|---|---|---|---|---|---|---|
| | Yes | | No | | OR | 95% CI | p-value | AOR | 95% CI | p-value |
| | n | % | n | % | | | | | | |
| Yes | 76 | 13.7 | 480 | 86.3 | 1.34 | 0.83–2.17 | 0.230 | | | |
| **Used to be greatly supported by parents in regard to clothes and other items while aged 0–5 years** | | | | | | | | | | |
| No | 23 | 23.0 | 77 | 77.0 | 1.00 | | | | | |
| Yes | 80 | 13.1 | 530 | 86.9 | 1.98 | 1.17–3.33 | 0.010* | | | |
| **Had accident while aged 0–5 years** | | | | | | | | | | |
| No | 75 | 12.7 | 516 | 87.3 | 1.00 | | | | | |
| Yes | 28 | 23.5 | 91 | 76.5 | 2.12 | 1.30–3.45 | 0.003* | | | |
| **Had been hospitalized while aged 0–5 years** | | | | | | | | | | |
| No | 72 | 14.4 | 429 | 85.6 | 1.00 | | | | | |
| Yes | 31 | 14.8 | 178 | 85.2 | 1.04 | 0.66–1.64 | 0.874 | | | |
| **Had head injury while aged 0–5 years** | | | | | | | | | | |
| No | 83 | 13.8 | 517 | 86.2 | 1.00 | | | | | |
| Yes | 20 | 18.2 | 90 | 81.8 | 1.38 | 0.81–2.37 | 0.235 | | | |
| **Had been physical assaulted by family member while aged 0–5 years** | | | | | | | | | | |
| No | 77 | 12.4 | 546 | 87.6 | 1.00 | | | 1.00 | | |
| Yes | 26 | 29.9 | 61 | 70.1 | 3.02 | 1.80–5.07 | <0.001* | 2.29 | 1.02–5.21 | 0.045* |
| **Had been physical assaulted by peer in school while aged 0–5 years** | | | | | | | | | | |
| No | 77 | 12.9 | 522 | 87.1 | 1.00 | | | | | |
| Yes | 26 | 23.4 | 85 | 76.6 | 2.07 | 1.58–3.42 | 0.004* | | | |
| **Major caregiver while aged 6–14 years** | | | | | | | | | | |
| Mother | 55 | 11.9 | 409 | 88.1 | 1.00 | | | | | |
| Father | 20 | 17.7 | 93 | 82.3 | 1.60 | 0.91–2.80 | 0.100 | | | |
| Stepfather | 3 | 42.9 | 4 | 57.1 | 5.58 | 1.21–25.58 | 0.027* | | | |
| Stepmother | 7 | 46.7 | 8 | 53.3 | 6.51 | 2.27–18.65 | <0.001* | | | |
| Relatives | 18 | 16.2 | 93 | 83.8 | 1.64 | 0.88–3.07 | 0.121 | | | |
| **Used to be greatly supported in regard to receiving desirable food and beverage from parents while aged 6–14 years** | | | | | | | | | | |
| No | 8 | 21.6 | 29 | 78.4 | 1.00 | | | | | |
| Yes | 95 | 14.1 | 578 | 85.9 | 1.68 | 0.75–3.78 | 0.211 | | | |
| **Had been greatly supported by parents to travel to desirable places while aged 6–14 years** | | | | | | | | | | |
| No | 15 | 25.9 | 43 | 74.1 | 1.00 | | | | | |
| Yes | 88 | 13.5 | 564 | 86.5 | 2.24 | 1.19–4.19 | 0.012* | | | |
| **Had been greatly supported by parents in regard to clothes and other items while aged 6–14 years** | | | | | | | | | | |
| No | 96 | 14.4 | 569 | 85.6 | 1.00 | | | | | |
| Yes | 7 | 15.6 | 38 | 84.4 | 1.09 | 0.47–2.52 | 0.837 | | | |
| **Had accident while aged 6–14 years** | | | | | | | | | | |
| No | 79 | 13.7 | 499 | 86.3 | 1.00 | | | | | |
| Yes | 24 | 18.2 | 108 | 81.8 | 1.14 | 0.85–2.32 | 0.186 | | | |
| **Had been hospitalized while aged 6–14 years** | | | | | | | | | | |
| No | 76 | 13.6 | 481 | 86.4 | 1.00 | | | | | |
| Yes | 27 | 17.6 | 126 | 82.4 | 1.36 | 0.84–2.19 | 0.214 | | | |

(*Continued*)

**Table 6.** (Continued)

| Factor | MA use | | | | Univariate analysis | | | Multivariate analysis | | |
|---|---|---|---|---|---|---|---|---|---|---|
| | Yes | | No | | OR | 95% CI | p-value | AOR | 95% CI | p-value |
| | n | % | n | % | | | | | | |
| **Had head injury while aged 6–14 years** | | | | | | | | | | |
| No | 78 | 13.1 | 517 | 86.9 | 1.00 | | | | | |
| Yes | 25 | 21.7 | 90 | 78.3 | 1.84 | 1.11–3.05 | 0.017* | | | |
| **Had been expelled from school while aged 6–14 years** | | | | | | | | | | |
| No | 92 | 13.3 | 601 | 86.7 | 1.00 | | | | | |
| Yes | 11 | 64.7 | 6 | 35.3 | 11.98 | 4.33–33.17 | <0.001* | | | |
| **Had been physically assaulted by family member while aged 6–14 years** | | | | | | | | | | |
| No | 80 | 12.3 | 570 | 87.7 | 1.00 | | | 1.00 | | |
| Yes | 23 | 38.3 | 37 | 61.7 | 4.43 | 2.50–7.84 | <0.001* | 3.15 | 1.32–7.54 | 0.010* |
| **Had been physically assaulted by peer in school while aged 6–14 years** | | | | | | | | | | |
| No | 81 | 13.0 | 541 | 87.0 | 1.00 | | | | | |
| Yes | 22 | 25.0 | 66 | 75.0 | 2.23 | 1.30–3.81 | 0.003* | | | |
| **Had been insulted due to sexual orientation while aged 6–14 years** | | | | | | | | | | |
| No | 97 | 14.2 | 584 | 85.8 | 1.00 | | | | | |
| Yes | 6 | 20.7 | 23 | 79.3 | 1.57 | 0.62–3.96 | 0.338 | | | |
| **Had been insulted due to socioeconomic status while aged 6–14 years** | | | | | | | | | | |
| No | 78 | 12.5 | 546 | 87.5 | 1.00 | | | | | |
| Yes | 25 | 29.1 | 61 | 70.9 | 2.87 | 1.70–4.84 | <0.001* | | | |
| **Was sexually abused while aged 6–14 years** | | | | | | | | | | |
| No | 96 | 13.8 | 601 | 86.2 | 1.00 | | | | | |
| Yes | 7 | 53.8 | 6 | 46.2 | 7.30 | 2.40–22.20 | <0.001* | | | |
| **Failed a class examination while aged 6–14 years** | | | | | | | | | | |
| No | 62 | 14.8 | 358 | 85.2 | 1.00 | 0.69–1.61 | 0.087 | | | |
| Yes | 41 | 14.1 | 249 | 85.9 | 1.05 | 0.69–1.61 | 0.087 | | | |
| **Number of close friends** (people) | | | | | | | | | | |
| ≤ 5 | 77 | 13.8 | 481 | 86.2 | 1.00 | | | | | |
| 6–10 | 23 | 16.3 | 118 | 83.7 | 2.12 | 0.73–2.02 | 0.447 | | | |
| ≥ 11 | 3 | 27.3 | 8 | 72.7 | 2.23 | 0.61–9.02 | 0.216 | | | |
| **Having a close friend who smokes** | | | | | | | | | | |
| No | 51 | 8.8 | 531 | 91.2 | 1.00 | | | | | |
| Yes | 52 | 40.6 | 76 | 59.4 | 7.12 | 4.52–11.23 | <0.001* | | | |
| **Having a close friend who drinks alcohol** | | | | | | | | | | |
| No | 57 | 11.1 | 458 | 88.9 | 1.00 | | | 1.00 | | |
| Yes | 46 | 23.6 | 149 | 76.4 | 2.48 | 1.61–3.81 | <0.001* | 2.42 | 1.24–4.72 | 0.009* |
| **Having a close friend who uses glue** | | | | | | | | | | |
| No | 96 | 13.8 | 599 | 86.2 | 1.00 | | | | | |
| Yes | 7 | 46.7 | 8 | 53.3 | 5.46 | 1.93–15.40 | 0.001* | | | |
| **Having a close friend who uses heroin** | | | | | | | | | | |
| No | 96 | 13.7 | 605 | 86.3 | 1.00 | | | | | |
| Yes | 7 | 77.8 | 2 | 22.2 | 22.06 | 4.52–107.75 | <0.001* | | | |

(*Continued*)

**Table 6.** (Continued)

| Factor | MA use | | | | Univariate analysis | | | Multivariate analysis | | |
|---|---|---|---|---|---|---|---|---|---|---|
| | Yes | | No | | OR | 95% CI | p-value | AOR | 95% CI | p-value |
| | n | % | n | % | | | | | | |
| **Having a close friend who uses MA** | | | | | | | | | | |
| No | 88 | 12.7 | 607 | 87.3 | 1.00 | | | | | |
| Yes | 15 | 100.0 | | 0.0 | 5.46 | 1.93–15.40 | 0.001* | | | |
| **Personality** | | | | | | | | | | |
| Polite and quiet | 25 | 4.0 | 601 | 96.0 | 1.00 | | | | | |
| Active and talkative | 54 | 90.0 | 6 | 10.0 | 0.96 | 0.58–1.59 | 0.865 | | | |
| Stays alone | 24 | 3.8 | 601 | 96.2 | 4.54 | 2.34–8.81 | <0.001* | | | |
| **Highly self-confident personality** | | | | | | | | | | |
| No | 20 | 9.5 | 190 | 90.5 | 1.00 | | | 1.00 | | |
| Yes | 83 | 16.6 | 417 | 83.4 | 1.89 | 1.23–3.17 | 0.016* | 2.35 | 1.17–4.69 | 0.016* |
| **Plays online games** | | | | | | | | | | |
| No | 27 | 9.1 | 271 | 90.9 | 1.00 | | | | | |
| Yes | 76 | 18.4 | 336 | 81.6 | 2.27 | 1.42–3.62 | 0.001* | | | |
| **Exercise regularly** | | | | | | | | | | |
| No | 24 | 26.7 | 66 | 73.3 | 1.00 | | | | | |
| Yes | 79 | 12.7 | 541 | 87.3 | 0.40 | 0.24–0.68 | 0.001* | | | |
| **Smokes** | | | | | | | | | | |
| No | 34 | 5.9 | 545 | 94.1 | 1.00 | | | 1.00 | | |
| Yes | 69 | 52.7 | 62 | 47.3 | 17.84 | 10.96–29.05 | <0.001* | 8.27 | 4.42–15.46 | <0.001* |
| **Uses alcohol** | | | | | | | | | | |
| No | 20 | 4.4 | 434 | 95.6 | 1.00 | | | | | |
| Yes | 83 | 32.4 | 173 | 67.6 | 10.41 | 6.20–17.50 | <0.001* | | | |
| **Used to use a "Facebook" application** | | | | | | | | | | |
| No | 3 | 6.5 | 43 | 93.5 | 1.00 | | | | | |
| Yes | 100 | 15.1 | 564 | 84.9 | 2.54 | 0.77–8.35 | 0.124 | | | |
| **Used to use a "Facebook" application** | | | | | | | | | | |
| No | 26 | 7.7 | 313 | 92.3 | 1.00 | | | | | |
| Yes | 77 | 20.8 | 294 | 79.2 | 3.15 | 1.97–5.06 | <0.001* | | | |
| **Used to work in a night-work sector** | | | | | | | | | | |
| No | 95 | 13.8 | 593 | 86.2 | 1.00 | | | | | |
| Yes | 8 | 36.4 | 14 | 63.6 | 3.57 | 1.46–8.73 | 0.005* | | | |
| **Used to have sex in exchange for items or money** | | | | | | | | | | |
| No | 99 | 14.1 | 601 | 85.9 | 1.00 | | | | | |
| Yes | 4 | 40.0 | 6 | 60.0 | 4.05 | 1.12–14.60 | 0.033* | | | |
| **Has been arrested** | | | | | | | | | | |
| No | 69 | 10.5 | 587 | 89.5 | 1.00 | | | | | |
| Yes | 34 | 63.0 | 20 | 37.0 | 14.46 | 7.89–26.51 | <0.001* | | | |
| **Knowledge on the impacts of MA use** | | | | | | | | | | |
| Low | 45 | 17.7 | 209 | 82.3 | 3.55 | 0.82–15.35 | 0.089 | | | |

(*Continued*)

**Table 6.** (Continued)

| Factor | MA use | | | | Univariate analysis | | | Multivariate analysis | | |
|---|---|---|---|---|---|---|---|---|---|---|
| | Yes | | No | | OR | 95% CI | p-value | AOR | 95% CI | p-value |
| | n | % | n | % | | | | | | |
| Moderate | 56 | 13.3 | 365 | 86.7 | 2.53 | 0.59–10.84 | 0.120 | | | |
| High | 2 | 5.7 | 33 | 94.3 | 1.00 | | | | | |
| **Attitude on the impacts of MA use** | | | | | | | | | | |
| Low | 48 | 22.7 | 163 | 77.3 | 4.42 | 2.02–9.68 | <0.001* | | | |
| Moderate | 47 | 12.7 | 324 | 87.3 | 2.18 | 1.00–4.74 | 0.005* | | | |
| High | 8 | 6.3 | 120 | 93.8 | 1.00 | | | | | |

*Significance level α = 0.05

physically assaulted by a family member while aged 0–5 years had a 2.29-fold (95% CI = 1.02–5.21) greater chance of MA use than those who had not. Those who had been physically assaulted by a family member while aged 6–14 years had a 3.15-fold (95% CI = 1.32–7.54) greater chance of MA use than those who had not. Those who had a close friend who used alcohol had a 2.24-fold (95% CI = 1.24–4.72) greater chance of MA use than those who did not. Those who had a highly confident personality had a 2.35-fold (95% CI = 1.17–4.69) greater chance of MA use than those who did not, and those who smoked had a 8.27-fold (95% CI = 4.42–15.46) greater chance of MA use than those who did not (Table 6).

## Discussion

Among the Akh and Lahu hill tribe youths who are living in northern Thailand, there was a high prevalence (14.5%) of MA use. There were also several factors related to MA use including personal characteristics, personality, family member and peer behaviors, and childhood experiences. Being male, being older, smoking, and having a highly confident personality were risk factors for using MA. Those who had a family member who used MA and had a close friend who used alcohol had a greater risk of using MA than those who did not. Childhood experiences of physical assault by a family member while aged 0–5 or 6–14 years were also associated with MA use among Akha and Lahu youths aged 15–24 years.

The prevalence of MA use among the youths who were studying in a vocational school in northern Thailand [27] was reported as 8.8%. This shows that the prevalence of MA use among the hill tribe youths (14.5%) is greater than that among the youths who were living in northern Thailand. It was also greater than the prevalence of MA use reported in Cambodia (10.4%) [28]. A greater proportion of MA use among the Akha and Lahu youths while comparing with Thai population, it could be supported by a qualitative study presented that social norms and also other positive personal perceptions among the Akha and Lahu were acting as major contributors for MA use in these population [24].

In our study, it was found that males had a significantly greater risk of using MA than females, which is consistent with a study conducted in Myanmar, which reported that males had a greater prevalence of MA users than females [29]. However, Dluzen et al [30] and Rung-nirundorn et al [31] reported that females were more likely to be MA users and significantly more likely to be MA-dependent than males. This might be because in the culture of the Akha and Lahu hill tribe people, males dominate all activities at the family and community levels; therefore, males could expose to and use MA more than females [24].

In this study, it was also found that people aged 18–20 years had a greater risk of using MA than the youngest Akha and Lahu youths. This could be because older youths have income from work, and they could afford to use MA. Moreover, older youths may have many more close friends from socializing, and the opportunities to begin using MA could be greater than those among younger youths. The World Health Organization (WHO), Thailand, reported that Thai youths experienced their first use of drugs before the age of 14 years [32]. A study in Malaysia in 2018 [33] also reported that the age of beginning MA use was 13 years, which supports our finding. A report from a national survey on drug use and health in the United States in 2015 also reported that the early age of MA use was 12 years [34]. However, a study in Australia in 2019 [35] reported that among youths in Australia, the first use of MA occurred at 20 years, which is different from our study.

Smoking was found to be associated with MA use among Akha and Lahu youth in Thailand. This finding was supported by a study in Thailand that reported that smoking was significantly associated with the initiation of MA use among youths [27]. A study in Morocco also reported that smoking behavior was associated with MA use among high school children [36]. Moreover, in a review of an epidemiologic study in 2016 [37], it was found that smoking behaviors were greatly associated with MA use. In a systematic review, it was presented that smoking was a major predictor for MA use in various age categories [38].

A highly self-confident personality was also found to be associated with MA use among Akha and Lahu youths in Thailand. This finding could be explained by the fact that those who have high confidence would have a chance to engage in a new experience in their life, particularly in the use of MA among the Akha and Lahu youths. Due to youths being in a stage of life in which they are very eager to know their environment along with access to MA and a low education, youths can become MA users. This concept is supported by studies conducted in Taiwan [39] and in the United States [40]. However, a study in Iran reported the idea that a highly self-confident personality type was a protective factor for MA use [41]. However, the Alcohol and Drug Abuse Institute (ADAI) reported that those who had low confidence had a greater risk of initiating MA use among American people [42].

Having a family member who uses MA was greatly significantly associated with MA use among Akha and Lahu youths in Thailand. This is supported by a study by Chomchoie et al. [24]. The systematic review study clearly showed an association between a family history of drug use and MS use among youths [38].

A study in the United States reported that physical abuse before 15 years of age was a key factor associated with MA use and MA-related violence [40]. Moreover, a study in Morocco reported that living with an unsecure family was associated with MA use among youths [36]. Peltzer et al. [43] demonstrated that being a victim of physical assault, particularly by family members, was associated with MA use among youths in Asia. In our study, it was found that those children had been physical assaulted either during age of 0–5 years or 6–14 years or both periods had a greater chance of MA use that those who did not. A total of 108 cases were reported in having physical assaulted by family member from whole participants; 87 cases reported on age of 0–5 years, and 60 cases were reported on aged of 6–14 years. Among the victims, 38 out of 109 cases (34.8%) had experienced on physical assaulted by family member in both periods.

This may be children in childhood need to get love and care from people living around them particularly from their parents and family members to grow strong both physical and mental health. Children who grew up with love and safe environment, it could motivate to get desired outcomes in later years of age such as not use MA [44, 45].

In our study, it was found that children those who had a close friend who used alcohol had a greater chance of MA use than those who did not. This is supported by a study in the United

Stated which was reported that those children who had a close friend who used alcohol had a greater chance to initiate MA than those who did not significantly [46]. A longitudinal study in rural cities, Wester United States, it was found that those adolescents who had a close friend who used alcohol was associated with substance use especially MA [47]. Moreover, a qualitative study in Thailand was also reported that having close friend who used alcohol led children to initiate MA [48].

Some limitations have been found in the study. First, identifying those people who used MA was difficult because it is an illegal substance in Thailand; therefore, most people who are using MA would not identify themselves as MA users. However, with the method of no information being traced back to any individual after filling in the questionnaire and the double-check method used by public health volunteers in a community to identify participants who used MA, the information gathered on the outcome would be closely related to the actual outcome. Under the current Thai government policy, all villagers have to be identified and classified in regard to whether they are using MA or not by their peers and an anonymous method. Those who are using MA are asked to participate in MA treatment programs in villages. This program is now working particularly well in rural villages and is managed by the Ministry of Public Health and other stakeholders [49]. Second, in part three of the questionnaire, questions were used to collect individual experience information related to when the participants were aged 0–5 years, particularly information related to physical assault by family members. These questions were answered by their parents, and the outcomes were shown to have high accuracy in the pilot test. Finally, three participants provided incomplete questionnaires, and they were excluded from the analysis. This small proportion of missing data would not interfere with the interpretation of the information.

## Conclusion

The study clearly shows the strong associations between childhood experiences while aged 0–14 years and personal behaviors and MA use among Akha and Lahu youths of older age in northern Thailand. Compared with other groups, male sex, smoking, older age, having close friends who use alcohol, and having a family member who uses MA were associated with MA use. Moreover, those who experienced physical assault from family members while aged 0–14 years were likely to use MA at a later age. Integrated intervention programs are urgently needed to reduce MA use among Akha and Lahu youths in Thailand; these programs should focus on improving family relationships and male individuals, smokers, and people with a highly confident personality. Moreover, the implementation should be focused in regularly monitoring and prevention on the physical assaulted during childhood by family members. The practical guideline on basic action while facing a problem of the physical assaulted in children by family members for the community health volunteers should be developed and provided. Strong collaborations among relevant agencies, both government and nongovernment, within countries and between counties are needed to address this problem.

## Supporting information

**S1 File. Questionnaire used in the study.**
(PDF)

**S2 File. Data file of the study.**
(XLSX)

## Acknowledgments

We would like to thank the National Research Council of Thailand (NRCT), Mae Fah Luang University (MFU), and The Center of Excellence for the Hill Tribe Health Research in support grant for doing this project. We also would like to extend our thank to community leaders and participants in their cooperation along the process of data collection.

## Author Contributions

**Conceptualization:** Tawatchai Apidechkul, Fartima Yeemard, Marisa Poomiphak Na Nongkhai, Rachanee Sunsern.

**Data curation:** Tawatchai Apidechkul, Chalitar Chomchoei, Pilasinee Wongnuch, Ratipark Tamornpark, Panupong Upala.

**Formal analysis:** Tawatchai Apidechkul, Chalitar Chomchoei, Ratipark Tamornpark.

**Funding acquisition:** Tawatchai Apidechkul.

**Investigation:** Tawatchai Apidechkul, Chalitar Chomchoei, Pilasinee Wongnuch, Ratipark Tamornpark, Panupong Upala, Fartima Yeemard, Marisa Poomiphak Na Nongkhai, Woottichai Nachaiwieng.

**Methodology:** Tawatchai Apidechkul, Marisa Poomiphak Na Nongkhai.

**Project administration:** Tawatchai Apidechkul.

**Supervision:** Pilasinee Wongnuch, Marisa Poomiphak Na Nongkhai, Woottichai Nachaiwieng, Rachanee Sunsern.

**Writing – original draft:** Tawatchai Apidechkul, Pilasinee Wongnuch, Ratipark Tamornpark, Panupong Upala, Fartima Yeemard, Marisa Poomiphak Na Nongkhai.

**Writing – review & editing:** Tawatchai Apidechkul, Chalitar Chomchoei, Marisa Poomiphak Na Nongkhai, Woottichai Nachaiwieng, Rachanee Sunsern.

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
