## [Decision Letter · Decision Letter 0]

13 Mar 2020

PONE-D-20-01893

Associations of childhood experiences and methamphetamine use among Akha and Lahu hill tribe youths in northern Thailand: A cross-sectional study

PLOS ONE

Dear Dr. Apidechkul,

Thank you for submitting your manuscript to PLOS ONE. After careful consideration, we feel that it has merit but does not fully meet PLOS ONE’s publication criteria as it currently stands. Therefore, we invite you to submit a revised version of the manuscript that addresses the points raised during the review process.

We would appreciate receiving your revised manuscript by Apr 27 2020 11:59PM. To enhance the reproducibility of your results, we recommend that if applicable you deposit your laboratory protocols in protocols.io, where a protocol can be assigned its own identifier (DOI) such that it can be cited independently in the future. For instructions see: http://journals.plos.org/plosone/s/submission-guidelines#loc-laboratory-protocols

We look forward to receiving your revised manuscript.

Kind regards,

Siyan Yi, MD, MHSc, PhD

Academic Editor

PLOS ONE

Journal Requirements:

3. Please address the following:

- Please include additional information regarding the survey or questionnaire used in the study and ensure that you have provided sufficient details that others could replicate the analyses. For instance, if you developed a questionnaire as part of this study and it is not under a copyright more restrictive than CC-BY, please include a copy, in both the original language and English, as Supporting Information.

- Please further explain how "individuals with a highly confident personality" were identified.

- Please provide additional details regarding participant consent. In the ethics statement in the Methods and online submission information, please ensure that you have specified (1) whether consent was informed and (2) what type you obtained (for instance, written or verbal, and if verbal, how it was documented and witnessed).

6. Your ethics statement must appear in the Methods section of your manuscript. If your ethics statement is written in any section besides the Methods, please move it to the Methods section and delete it from any other section. Please also ensure that your ethics statement is included in your manuscript, as the ethics section of your online submission will not be published alongside your manuscript.

Reviewers' comments:

Reviewer's Responses to Questions

**Comments to the Author**

1. Is the manuscript technically sound, and do the data support the conclusions?

Reviewer #1: Yes

Reviewer #2: Partly

Reviewer #3: Yes

Reviewer #4: Partly

2. Has the statistical analysis been performed appropriately and rigorously? 

Reviewer #1: Yes

Reviewer #2: No

Reviewer #3: Yes

Reviewer #4: No

3. Have the authors made all data underlying the findings in their manuscript fully available?

Reviewer #1: Yes

Reviewer #2: No

Reviewer #3: No

Reviewer #4: No

4. Is the manuscript presented in an intelligible fashion and written in standard English?

Reviewer #1: Yes

Reviewer #2: Yes

Reviewer #3: Yes

Reviewer #4: Yes

5. Review Comments to the Author

Reviewer #1: The work aimed to determine the factors associated with amphetamine use among the hill tribe youths by a cross-sectional among 710 Akha and Lahu youths.

A. More clarification on the accuracy of gathering information on MA use is needed. Validity of instrument?

B. Provide reason of collecting data on violence during 0-5 years from parents.

Reviewer #2: The authors survey 2 population (as in sampling method) but presented in all which is inappropriate. Akha and Lahu are different ethnic group, thus cannot combine in analysis (could be compared).

Methodology such as sampling, sample size determination, Tool development are skeptical.

It seems that the findings are not significant related to Akha and Lahu culture specifically (it could be found in any youth in Thailand and other countries)

Reviewer #3: Comments from reviewers for the paper entitled “Associations of childhood experiences and methamphetamine use among Akha and Lahu hill tribe youths in northern Thailand: A cross-sectional study”

Abstract:

Should rewrite the result part more shortly as the authors wrote the result of the multivariate analysis quite details. Recommendation should be added by addressing the physical assault.

Introduction

Please add the prevalence of MA use in the Chiang Rai province, especially among this hill tribe.

The two sentences after references [14] should be combined into one sentence as the sentence is too short.

Please explain the term of childhood experiences. Please also add some references related to childhood experiences and MA use.

Methods

Authors put the heading Study population, eligible, Inclusion and exclusion criteria then study sample, which are confusing. Please put only Study sample with inclusion and exclusion criteria. Then the authors could describe about sampling, how the authors did randomly sampling, please describe more detail which method of randomly sapling used? Especially authors mentioned 496 people from 10 Akha villages and 518 people from 10 Lahu villages, was it random sampling or opportunistic

For sample size, please add the reference of p=0.27.

Researchers developed the Instruments based on the previous instruments or not, if yes, please give the references such as instruments related to chilhood experiences. This is interesting that the authors got the information from parents which had recall bias, how long that the authors asked back in the childhood experiences. Did the authors also asked the children themselves, to cross-check the information of childhood experiences. Please give the references related to the Instruments of history of being assaulted, history of school expulsion, assault by family members, especially assault by peers, did the parents knew about these issues.

Please also add the references of the questions related to knowledge and attitude toward MA use.

For the instrument, please specific which information that collected from parents and which information collected from children? As I am wondering that the authors collected the socio0denograhic of parents from themselves or collected from children. As the authors mentioned only that they collected childhood information from parents.

Please indicate IOC value and Cronbach’s alpha of 0.78 of which questions.

Pilot test should be conducted at least 30 participants, but the authors conducted pilot text only with 20, especially the authors had different target group Akha and other hill tribes, they should have at least 30 for each tribe.

Data Collection

Please specify which method of randomly selected the villages such as the simple, systematic and tec..

How select the participants from each village as the authors mentioned “496 people from 10 Akha villages and 518 people from 10 Lahu villages” Did the authors used PPS. Then how the authors did selected the participants from each selected village? Please specify the specific method of random sampling. Which methods that the authors interviewed the participants? How did you ensured the participants with low or not literacy? There were 71 participants had no education, how the participants answered to the questionnaire. What is the response rate?

Ethical

Please indicate the ethical approval Number, and how did the authors get the consent form from both parents and children.

Statistical Analysis

Please describe more about the multivariate analysis such as multiple logistic regression, which methods used, and which variables were enetred into the final model. Need to describe your statistical modeling techniques in much more detail.

Result

Table 2. History of family members’ exposure to drugs and alcohol

Please change to be Ever had or having as the authors included the past and currently.

Table 5: Participants’ behaviors and personality

Used MA at least once should be changed to be “Ever used MA at least once” and this should be included all variables as Ever used included past and currently.

Please indicate which variables entered in the model and the good fitness model. Please present the OR and 95%CI in the result for the variables significantly associated wit MA use. The authors showed in the table 6, but they did not described in the text.

Table 6 is too long, the authors could put in the appendix. The authors could briefly describe which variables were entered into the final model and which p valued in the univariate analysis, then the authors used which method for multivariate analysis such as backward elimination and presented only the final model that were significantly associated with MA use.

Discussion

This part is too short and please expand the discussion according to the variables were significantly associated wih MA use such as friends drinking related to MA? How/ Why? Had been physical assaulted by family member during childhood while aged 0-5 years and 6-14 years? How this related to MA? Are there any participants have been assault both during the 2 occasions. Please add more discussions and references to support for the factors associated with MA use.

Recomemndation should be revised according to the findings such as prevention of physical assaulted by family member during childhood. Please give more specific recomemndations.

Grammar - I would recommend reading through the manuscript purely to identify grammatical errors and awkward phrasing.

Reviewer #4: General comments:

This paper addressed an important topic and one of special relevance in Thailand where the methamphetamine (MA) epidemic has a profound impact on communities, especially in the north of the country. The paper presents results of a cross-sectional study evaluating the association of childhood experiences and the use of MA among hill tribe youth in northern Thailand. This is an important addition to the literature on the topic and the results can inform public health action and intervention in the area. The paper is mostly descriptive and the statistical methods could be explained in more detail. The paper could benefit from careful editing both for language and for clarity and minimizing logical repetitions.

Specific comments:

1. The abstract methods section could be edited and shortened to avoid repetitions of the phrase “…greater risk of MA use than those who did not…”

2. In the abstract, conclusions: the last sentence and intervention for “male youth” need to be better explained (I suppose as a target population)

3. Inclusion and exclusion: seems like the only criteria was age and the affiliation with the tribes, and understanding of Thai. And it will be helpful to note early on the population included youth who live in the tribal villages in the north of the country.

4. Sample size calculation: for a comparison of two independent proportions, in addition to a proportion for the one reference group a difference in proportion and the power level are needed to complete the calculations. Please add the necessary details. Actually, reading further, the analysis does not compare the two samples at all – rather, it is the association with MA use. So, the sample size need to reflect this!

5. Research instrument section: Survey questions in Part 3 on childhood abuse in ages 0-5 refer also to abuse by family members, yet it is noted that this part was filled in by parents?? Why would this be done? Also for those that are >18?

6. Piloting the questionnaire: were modification done between repeated cycles of the pilot? And what data was used for the reliability assessment?

7. Sample: after the 10 villages were randomly selected – how were the individual sampled? Were all youth in each village approached? Also more than one youth per household? In which case the analysis need to take this into account.

8. Results: what wasa the response rate? And what were reasons for non-participation?

9. Table 1: may be interesting to add two columns comparing the two sample, in addition to the total. Also, factors that were mostly missing such as income could be deleted from the tables

10. Tables 2-5 : can some of this information be presented in graphical form? Some items, for example “social behavior: yes/no” are hard to interpret; similarly items such as “accident” , “hospitalization” etc. are all these factors relevant? Also check the yes/no frequencies for question on parents support.

11. Table 6: to reduce length, possibly delete variables with p-value >0.2 or 0.3. also , some factors with small cells could be combined e.g. friends who use glue/heroin etc

12. How was the MV model developed? The initial model (all variables presented in table 6 presumably) have a mixture of socio-demographic characteristic, exposures and self substance use, other risk factors including childhood abuse but also type/circumstance of MA use. A more careful consideration of what is relevant to include in the model may be useful. In addition, How was the final MV model determined? Stepwise procedure? Showing only those factors that were significant (wrong approach)?

13. Was tribe a modifying factor for some covariates? The N may be large enough to allow some more refined analysis

6. PLOS authors have the option to publish the peer review history of their article (what does this mean?). If published, this will include your full peer review and any attached files.

Reviewer #1: Yes: Karl Peltzer

Reviewer #2: No

Reviewer #3: No

Reviewer #4: No

---

## [Author Response · Author response to Decision Letter 0]

13 Apr 2020

PONE-D-20-01893

Associations of childhood experiences and methamphetamine use among Akha and Lahu hill tribe youths in northern Thailand: A cross-sectional study

PLOS ONE

Dear Dr. Apidechkul,

Thank you for submitting your manuscript to PLOS ONE. After careful consideration, we feel that it has merit but does not fully meet PLOS ONE’s publication criteria as it currently stands. Therefore, we invite you to submit a revised version of the manuscript that addresses the points raised during the review process.

We would appreciate receiving your revised manuscript by Apr 27 2020 11:59PM. To enhance the reproducibility of your results, we recommend that if applicable you deposit your laboratory protocols in protocols.io, where a protocol can be assigned its own identifier (DOI) such that it can be cited independently in the future. For instructions see: http://journals.plos.org/plosone/s/submission-guidelines#loc-laboratory-protocols

• A rebuttal letter that responds to each point raised by the academic editor and reviewer(s). This letter should be uploaded as separate file and labeled 'Response to Reviewers'.

• A marked-up copy of your manuscript that highlights changes made to the original version. This file should be uploaded as separate file and labeled 'Revised Manuscript with Track Changes'.

• An unmarked version of your revised paper without tracked changes. This file should be uploaded as separate file and labeled 'Manuscript'.

We look forward to receiving your revised manuscript.

Kind regards,

Siyan Yi, MD, MHSc, PhD

Academic Editor

PLOS ONE

Journal Requirements:

: The links provided are not working, however I have checked and improved according to the journal styles.

 : It has been done and improved by AJE, with verification code: 6EAB-88FC-7B3F-EF75-D4F1 . 

: Certification attached.

3. Please address the following:

- Please include additional information regarding the survey or questionnaire used in the study and ensure that you have provided sufficient details that others could replicate the analyses. For instance, if you developed a questionnaire as part of this study and it is not under a copyright more restrictive than CC-BY, please include a copy, in both the original language and English, as Supporting Information.

- Please further explain how "individuals with a highly confident personality" were identified.

: Improved

- Please provide additional details regarding participant consent. In the ethics statement in the Methods and online submission information, please ensure that you have specified (1) whether consent was informed and (2) what type you obtained (for instance, written or verbal, and if verbal, how it was documented and witnessed).

 : Improved

: Data has been uploaded. 

: 0000-0001-8301-2055

6. Your ethics statement must appear in the Methods section of your manuscript. If your ethics statement is written in any section besides the Methods, please move it to the Methods section and delete it from any other section. Please also ensure that your ethics statement is included in your manuscript, as the ethics section of your online submission will not be published alongside your manuscript.

: Improved

Reviewers' comments:

Reviewer's Responses to Questions

Comments to the Author

1. Is the manuscript technically sound, and do the data support the conclusions?

Reviewer #1: Yes

Reviewer #2: Partly

Reviewer #3: Yes

Reviewer #4: Partly

2. Has the statistical analysis been performed appropriately and rigorously? 

Reviewer #1: Yes

Reviewer #2: No

Reviewer #3: Yes

Reviewer #4: No

3. Have the authors made all data underlying the findings in their manuscript fully available?

Reviewer #1: Yes

Reviewer #2: No

Reviewer #3: No

Reviewer #4: No

4. Is the manuscript presented in an intelligible fashion and written in standard English?

Reviewer #1: Yes

Reviewer #2: Yes

Reviewer #3: Yes

Reviewer #4: Yes

5. Review Comments to the Author

Reviewer #1: The work aimed to determine the factors associated with amphetamine use among the hill tribe youths by a cross-sectional among 710 Akha and Lahu youths.

A. More clarification on the accuracy of gathering information on MA use is needed. Validity of instrument?

: Improved in section of “method”

B. Provide reason of collecting data on violence during 0-5 years from parents.

: Improved, thank you for the comment.

Reviewer #2: The authors survey 2 population (as in sampling method) but presented in all which is inappropriate. Akha and Lahu are different ethnic group, thus cannot combine in analysis (could be compared).

Methodology such as sampling, sample size determination, Tool development are skeptical.

It seems that the findings are not significant related to Akha and Lahu culture specifically (it could be found in any youth in Thailand and other countries)

: Thank you for the comment, I (Dr.Tawatchai Apidechkul) as the principle investigator have long experience in working with the hill tribe in Thailand for more than 15 years. I really known that the hill tribes have different cultures, some aspect very big different, but some aspect is not too big. However, understanding in practice for MA use, I feel that it is not too much relevant to the tribe cultures but rather in parenting styles. The original idea for doing this project was from our observation on the date we visited the village, we found that some youths have started in use of MA while the other did not. This scenario was similar all tribes. But the reason to do for two tribes, because of research budget and also other components such as the feasibility to test the research hypothesis. With the help of WHO-Thailand staff and also the staff of Harvard Medical School who are my advisor. Then the research was completed and the findings come out. 

We had done a small project before doing this study, please see deatil: BMC Public Health, 2019. DOI: 10.1186/s12889-019-7226-y 

Reviewer #3: Comments from reviewers for the paper entitled “Associations of childhood experiences and methamphetamine use among Akha and Lahu hill tribe youths in northern Thailand: A cross-sectional study”

Abstract:

Should rewrite the result part more shortly as the authors wrote the result of the multivariate analysis quite details. Recommendation should be added by addressing the physical assault.

: Thank you very much for the comment. It is improved.

Introduction

Please add the prevalence of MA use in the Chiang Rai province, especially among this hill tribe.

: Thank you for the comment. I have tries so many times to seek the Thai and English information on the prevalence of MA use among people in Chiang Rai and also in the hill tribe, unfortunately , there is no information available. This was the information that we have had investigated before, during the study including in the period of writing up our manuscript, but no scientific or event the government’s report on the situation of the MA use among the hill tribe people. 

The two sentences after references [14] should be combined into one sentence as the sentence is too short.

: Improved.

Please explain the term of childhood experiences. Please also add some references related to childhood experiences and MA use.

:Thank you for the comment. Based on our study, we just interest in detecting the association of individuals’ experience and MA use in later of their life among the hill tribe youths who are living in poor education and economic status. 

UNICEF [United Nation International Children’s Emergency Fund. Childhood under threat : The state of the world’s children 2005. Available from: https://www.unicef.org/sowc05/english/childhooddefined.html ] defines the childhood as the time of children to be in school and at play, to grow strong and confident with the love and encouragement of their family and an extended community of caring adults. Therefore the childhood experience is the experience of children during their childhood period which could be positive and negative impact on their later life. 

Since, the purpose of the study and also the design of the study were not focused on detection the association of neither negative or positive experience during childhood, so called “adverse Childhood experience (ACEs)” [Centers for Disease Control and Prevention (CDC). Violence prevention: adverse childhood experience (ACEs). Available from: https://www.cdc.gov/violenceprevention/childabuseandneglect/acestudy/index.html] and MA use, but we did a cross-sectional by assessing many exposures (including childhood experience) and MA use, then we avoid to use the word of “Adverse childhood experience” in the study. 

Therefore, with the different conditions of the child experience (which is one of the exposures) and ACEs, then we decide to not put the ACEs into the paper, to avoid the confusion of readers. We very hope you understand us.

Methods

Authors put the heading Study population, eligible, Inclusion and exclusion criteria then study sample, which are confusing. Please put only Study sample with inclusion and exclusion criteria. Then the authors could describe about sampling, how the authors did randomly sampling, please describe more detail which method of randomly sapling used? Especially authors mentioned 496 people from 10 Akha villages and 518 people from 10 Lahu villages, was it random sampling or opportunistic

For sample size, please add the reference of p=0.27.

: It is referred to the reference no. 26 which is presented from the beginning. [Toeam A, Lapvongwatana P, Chansatitporn N, Chamroonsawasdi K. Predictive Factors of Amphetamine Use Among Youths in a Congested Community. Thai Red Cross Nursing Journal. 2016; 9(2): 88-103] 

Researchers developed the Instruments based on the previous instruments or not, if yes, please give the references such as instruments related to childhood experiences. This is interesting that the authors got the information from parents which had recall bias, how long that the authors asked back in the childhood experiences. Did the authors also asked the children themselves, to cross-check the information of childhood experiences. Please give the references related to the Instruments of history of being assaulted, history of school expulsion, assault by family members, especially assault by peers, did the parents knew about these issues.

Please also add the references of the questions related to knowledge and attitude toward MA use.

: We had done a qualitative approach in understanding the MA use among these two population [Chomchoei C, Apidechkul T, Wongnuch P, Tamorapark R, Upala P, Nongkhai MP. Perceived factors influemcing the initiation of methamphetamine use among Akha and Lahu youths: a qualitative approach. BMC Public Health. 2019; 19: 847: DOI: 10.1186/s12889-019-7226-y.] before doing this project. We used the information from our qualitative phase to develop the tool especially the questionnaire in this quantitative phase. 

: The questions related to history of being assaulted, history of school expulsion, assault by family members, especially assault by peers were also obtained from our own experience in doing research in these populations, literature review, and also consulted with expert. We did not sure that the parents known about the issues. Thank you very much for pointing out the great issue. We will discussion with team to find the great method in our next project to solve the problem in family and community levels. 

: Along the steps of doing the project, parents were not informed anything regarding their child. And we did not do the cross-check information between parents and child information on 0-5 years because from our pilot phase, we did check the accuracy and found that their provided mostly similar between parents and child. Moreover, after the questionnaire reaching to researchers, no information could reflect to any individual’s information, then we did not do cross-check between parent and child during the study. 

For the instrument, please specific which information that collected from parents and which information collected from children? As I am wondering that the authors collected the socio0denograhic of parents from themselves or collected from children. As the authors mentioned only that they collected childhood information from parents.

: Yes, only part three which was asking the child experience during aged 0-5 years, asked from their parents to improve the quality of the information due to the participants (children) were very young. In questionnaire part two which is asking on socioeconomic status was completed by participants (children). This is supported from the pilot period that most of the participants (children) have much more fluent in Thai (questionnaire was provided in Thai) than their parents and more understand the question. 

Please indicate IOC value and Cronbach’s alpha of 0.78 of which questions.

Pilot test should be conducted at least 30 participants, but the authors conducted pilot text only with 20, especially the authors had different target group Akha and other hill tribes, they should have at least 30 for each tribe.

: Thank you for the comment. The IOC method is normally done by three experts in the field which is the common method. In pilot phase, we did for 10 participants because several reasons; 1) we have had some information from our previous study (reference no. 24); 2) to get the participants who were using Ma was very difficult; and 3) in the process of detecting the Cronbach’s alpha of 0.78, we did 3 times which is enough to improve the quality of the questions in section of KAP. 

Data Collection

Please specify which method of randomly selected the villages such as the simple, systematic and tec..

: Improved

How select the participants from each village as the authors mentioned “496 people from 10 Akha villages and 518 people from 10 Lahu villages” Did the authors used PPS. Then how the authors did selected the participants from each selected village? Please specify the specific method of random sampling. Which methods that the authors interviewed the participants? How did you ensured the participants with low or not literacy? There were 71 participants had no education, how the participants answered to the questionnaire. What is the response rate?

: After making simple random method to get the selected list of the villages, people aged 15-24 who were living in the villages were invited to participate the study. The response rate was was 77.6% ( 385 of 496) in Akha, and 62.7% (325 of 518) in Lahu. This were acceptable for the survey design with some sensitive problem as MA. After getting the response, we had discussed on the response rates, but we have found that there were not different from our previous works among people in these two tribes. 

: Those who responded on no-educated but still able to complete the questionnaire. This is also not surprise for us because from our previous studies and also our prior study in MA (reference No.24), we found that those who did not completely complete the 6th graded in primary school (grade 1-6), they would prefer to answer no education. For instance, a person finished in the 4th graded of primary school, with the answer of the questionnaire provided; no-education, primary education, secondary school, high school, vocational school, and university degree. They will answer in “no-educate”. The answer provided in the questionnaire is ordinal scales with one possible choice in each participant. 

Ethical

Please indicate the ethical approval Number, and how did the authors get the consent form from both parents and children.

: Improved

Statistical Analysis

Please describe more about the multivariate analysis such as multiple logistic regression, which methods used, and which variables were enetred into the final model. Need to describe your statistical modeling techniques in much more detail.

:Improved

Result

Table 2. History of family members’ exposure to drugs and alcohol

Please change to be Ever had or having as the authors included the past and currently.

: Improved and thank you very much.

Table 5: Participants’ behaviors and personality

Used MA at least once should be changed to be “Ever used MA at least once” and this should be included all variables as Ever used included past and currently.

:Improved and thank you very much.

Please indicate which variables entered in the model and the good fitness model. Please present the OR and 95%CI in the result for the variables significantly associated wit MA use. The authors showed in the table 6, but they did not described in the text.

: Thank you very much for the comment. In the step of analysis, we started in consideration in each group of independent variables with dependent variable according to the conceptual framework. Along the analysis, pseudo R2 of Cox-Snell R2 and Nagelkerke R2 were used for the determination of fit of the model, and chi-square of the model was used for determination of the prediction of model to the dependent variable. In each step, those predicting variables which was not significant in the model, were excluded from the model because the statistic on fitting model and predicting the model showed not good to fit the model. In the best fit model in the last model (multivariate model) was used for presenting the most fit model in explaining the associations. And in the final step after 8 variables presented the associations, we have controlled the impact of “tribe, marital status, religion, education, occupation, and Thai ID card” in the final model to fit the associations.

: This is why we are not presenting every variables in the model, because we need to find the best fit model to explain the association according to both epidemiological and biostatistics concepts.

Table 6 is too long, the authors could put in the appendix. The authors could briefly describe which variables were entered into the final model and which p valued in the univariate analysis, then the authors used which method for multivariate analysis such as backward elimination and presented only the final model that were significantly associated with MA use.

: As this is the first pioneer study, we would like to show all variables that much related to the MA use. However, many variables, which found not too much relevant to the MA, have been deleted from the table. Since this is the fisrt pioneer study on MA among the two major hill tribe youths in Thailand, we very hope that many variables will be considered for the next research study. 

: In the step of the analysis, we used “ENTER” mode which is allowed the researcher to consider the association in every step and we used the statistic on identifying in fitting the model. We did not concern only the association of the statistics but we also consider on the the association of the public health. Moreover, we used our original conceptual framework for doing the analysis. 

: In the step of analysis, we started in consideration in each group of independent variables with dependent variable according to the conceptual framework. Along the analysis, pseudo R2 of Cox-Snell R2 and Nagelkerke R2 were used for the determination of fit of the model, and chi-square of the model was used for determination of the prediction of model to the dependent variable. 

Discussion

This part is too short and please expand the discussion according to the variables were significantly associated wih MA use such as friends drinking related to MA? How/ Why? Had been physical assaulted by family member during childhood while aged 0-5 years and 6-14 years? How this related to MA? Are there any participants have been assault both during the 2 occasions. Please add more discussions and references to support for the factors associated with MA use.

: Thank you very much for the great question. There are 38 cases that presented had been physical assaulted in both period of life. 

: We have extended our discussion to cover all aspects as suggested with numbers of references. Thank you very much for great comments here.

Recomemndation should be revised according to the findings such as prevention of physical assaulted by family member during childhood. Please give more specific recomemndations.

: Thank you for the comment, it’s improved. 

Grammar - I would recommend reading through the manuscript purely to identify grammatical errors and awkward phrasing.

: Thank you very much, it is checked by the American Journal Experts with code no. 6EAB-88FC-7B3F-EF75-D4F1 . 

Reviewer #4: General comments:

This paper addressed an important topic and one of special relevance in Thailand where the methamphetamine (MA) epidemic has a profound impact on communities, especially in the north of the country. The paper presents results of a cross-sectional study evaluating the association of childhood experiences and the use of MA among hill tribe youth in northern Thailand. This is an important addition to the literature on the topic and the results can inform public health action and intervention in the area. The paper is mostly descriptive and the statistical methods could be explained in more detail. The paper could benefit from careful editing both for language and for clarity and minimizing logical repetitions.

Specific comments:

1. The abstract methods section could be edited and shortened to avoid repetitions of the phrase “…greater risk of MA use than those who did not…”

: Thank you very much, it was improved. 

2. In the abstract, conclusions: the last sentence and intervention for “male youth” need to be better explained (I suppose as a target population)

: Improved

3. Inclusion and exclusion: seems like the only criteria was age and the affiliation with the tribes, and understanding of Thai. And it will be helpful to note early on the population included youth who live in the tribal villages in the north of the country.

: Improved

4. Sample size calculation: for a comparison of two independent proportions, in addition to a proportion for the one reference group a difference in proportion and the power level are needed to complete the calculations. Please add the necessary details. Actually, reading further, the analysis does not compare the two samples at all – rather, it is the association with MA use. So, the sample size need to reflect this!

: Thank you for the comment. As we used a cross-sectional design which aimed to estimate the overall of the prevalence of MA use and to determine the factors associated with MA use. We did not aim to make a comparison from the earlier. Then, during the sample size calculation, it was looked as a whole sample size of the study sample. However, to much more sense in interpreting the finding, we divided into two proportional groups. 

: Based on the concept of epidemiological studies in a cross-sectional design, two or more groups is not allowed. We measures both many exposures and outcome (MA use) in the same point of time according to its design. 

5. Research instrument section: Survey questions in Part 3 on childhood abuse in ages 0-5 refer also to abuse by family members, yet it is noted that this part was filled in by parents?? Why would this be done? Also for those that are >18?

: Yes, this section (part 3) was asked on the experience during 0-5 years on the participants, but from our pilot phase we have found that the most abest way to gather the most accuracy information was from their parents due to the recall ability of participants (children). Therefore, we have asked these questions (part no3) from parents instead from children. 

6. Piloting the questionnaire: were modification done between repeated cycles of the pilot? And what data was used for the reliability assessment?

: Yes, during the repeated cycles, many point had been improved such as the sentences, words, phrases used in the questionnaire, and also the order of the questions. Some words are very difficult and not familiar in the Akha and Lahu circumstance.

: The reliability test, we detected on KAP and present in overall Cronbach alpha of the questionnaire in the last cycle. Because in the first and second cycles, we focus on the feasible and understanding of the questions used, word used, paragraph used and the order of the questions. 

7. Sample: after the 10 villages were randomly selected – how were the individual sampled? Were all youth in each village approached? Also more than one youth per household? In which case the analysis need to take this into account.

: Yes, all youths in the selected villages who met the criteria were invited to join the study. No conditions on the number of children in a household. After we got the list from village headman, we approach everyone. 

8. Results: what wasa the response rate? And what were reasons for non-participation?

: Thank you for the comment, it was provide the response rate in section of the results.

9. Table 1: may be interesting to add two columns comparing the two sample, in addition to the total. Also, factors that were mostly missing such as income could be deleted from the tables

: Improved

10. Tables 2-5 : can some of this information be presented in graphical form? Some items, for example “social behavior: yes/no” are hard to interpret; similarly items such as “accident” , “hospitalization” etc. are all these factors relevant? Also check the yes/no frequencies for question on parents support.

: Thank you for the comments. Even this is a good idea, however, based on our raw data to present in form of graphic is very difficult. 

: We so sorry, after getting back from AJE who checked and corrected the grammar for the paper, we did not checking again before submitting to the journal. In terms of “social behavior”, we mean socialized behavior. We have found that those who have much more activities including joining in parties were more at risk in use of MA. This is the original in putting the question into the questionnaire. Since this variable is a poor predictor in the logistic regression model, it was deleted from table no.6

11. Table 6: to reduce length, possibly delete variables with p-value >0.2 or 0.3. also , some factors with small cells could be combined e.g. friends who use glue/heroin etc

: Thank for the great comments. We have tried to deleted some variables that not too much related to the MA use such as p-value > 0.3 and did cell combination in some variables as suggestions which are related information in other tables such as table 1-4.

12. How was the MV model developed? The initial model (all variables presented in table 6 presumably) have a mixture of socio-demographic characteristic, exposures and self substance use, other risk factors including childhood abuse but also type/circumstance of MA use. A more careful consideration of what is relevant to include in the model may be useful. In addition, How was the final MV model determined? Stepwise procedure? Showing only those factors that were significant (wrong approach)?

: Thank you very much for the comment. Actually, we used “Enter” mode to extract the final model based on the conceptual framework in a cross-sectional method. Then to use “Backward” or “Forward” or “Stepwise” is not a good technique since we have to consider not just statistical significant but also public health significant (marginal significant) but it is needed to be included into the model. Under some condition, we have to consider on the size of the association (OR) and confident interval (CI), even the p-value is not shown the significance. 

In the step of analysis, we started in consideration in each group of independent variables with dependent variable according to the conceptual framework. Along the analysis, pseudo R2 of Cox-Snell R2 and Nagelkerke R2 were used for the determination of fit of the model, and chi-square of the model was used for determination of the prediction of model to the dependent variable. 

In the final step after 8 variables presented the associations, we have controlled the impact of “tribe, marital status, religion, education, occupation, and Thai ID card” in the final model to fit the associations.

13. Was tribe a modifying factor for some covariates? The N may be large enough to allow some more refined analysis

: Thank you very much for the suggestion. We have planned to do that, however, the analysis and results presented this article is based on our original purpose. We will do for analysis in different tribe, if any interest comes will be find a proper place for publication.

Thank you very much!

TK

---

## [Decision Letter · Decision Letter 1]

7 May 2020

PONE-D-20-01893R1

Associations of childhood experiences and methamphetamine use among Akha and Lahu hill tribe youths in northern Thailand: A cross-sectional study

PLOS ONE

Dear Dr. Apidechkul,

Thank you for submitting your manuscript to PLOS ONE. After careful consideration, we feel that it has merit but does not fully meet PLOS ONE’s publication criteria as it currently stands. Therefore, we invite you to submit a revised version of the manuscript that addresses the points raised during the review process.

We would appreciate receiving your revised manuscript by Jun 21 2020 11:59PM. To enhance the reproducibility of your results, we recommend that if applicable you deposit your laboratory protocols in protocols.io, where a protocol can be assigned its own identifier (DOI) such that it can be cited independently in the future. For instructions see: http://journals.plos.org/plosone/s/submission-guidelines#loc-laboratory-protocols

We look forward to receiving your revised manuscript.

Kind regards,

Siyan Yi, MD, MHSc, PhD

Academic Editor

PLOS ONE

Reviewers' comments:

Reviewer's Responses to Questions

**Comments to the Author**

1. If the authors have adequately addressed your comments raised in a previous round of review and you feel that this manuscript is now acceptable for publication, you may indicate that here to bypass the “Comments to the Author” section, enter your conflict of interest statement in the “Confidential to Editor” section, and submit your "Accept" recommendation.

Reviewer #2: All comments have been addressed

Reviewer #4: (No Response)

2. Is the manuscript technically sound, and do the data support the conclusions?

Reviewer #2: No

Reviewer #4: Partly

3. Has the statistical analysis been performed appropriately and rigorously? 

Reviewer #2: No

Reviewer #4: No

4. Have the authors made all data underlying the findings in their manuscript fully available?

Reviewer #2: No

Reviewer #4: No

5. Is the manuscript presented in an intelligible fashion and written in standard English?

Reviewer #2: Yes

Reviewer #4: Yes

6. Review Comments to the Author

Reviewer #2: Although the author respond to comments, it is unclear.

Sampling method was straified cluster sampling, thus sample size calculation needs adjustment for design effect.

Reliability process is unusal (three rounds with the same 20 samples and calulated alpha from last round). Recall bias appeared. Alpha should be presented for individual section rather than overall.

Table 2-5 should be compare between two hill-tribes.

Although the MA lifetime prevalence is 3 times higher than general population of Thailand in 2019, it should be dicussed why it is.

The specific culture of these hill-tribes affected to MA use still need to clarify.

Reviewer #4: The authors were responsive to comments and the paper is now improved and has greater clarity overall. However, not all the relevant responses that are given in the letter to the editor are included in the paper. I have a few remaining comments:

Specific comments:

1. In the abstract, conclusions: the last sentence and intervention for “male youth” need to be better explained (I suppose as a target population) – this sentence is still not well phrased. is the intention to say: …interventions that lowers risk of MA use by addressing family relationship, male youth risk behaviors….etc….?

2. Sample size calculation: still need clearer description of what was done under what assumption. if the calculations were to estimate the proportion of MA use, then a single sample with confidence interval widths could be used. if it is based on comparing two proportions (which is what is indicated), then, explain which groups are compared and if this is within each tribe group? - this may not be a critical point in the paper but if included, it should be clear and relevant to the analysis performed.

3. As more than one youth per household could participate, can the authors add the info on how many clusters with size>1 were included? as the analysis does not take this clustering into account at a minimum, a note referring to potential biases in estimating the statistical significance (standard errors of estimates) should be included in methods and/or discussion. this is an important methodological issue.

4. Results: I do not find the reported “response rate” to the survey in the manuscript.

5. Table 1-5: the paper would benefit by creating tables 1-5 that contain more information including: a) the original n & 5 columns for overall frequencies and combining with it the univariate analysis that appears in table 6. then table 6 would just have the final multivariate table. this will be make it easier for the reader to capture the final model.

6. Description on how the MV model developed should be added to methods (it is in the response letter but not in the paper)

7. for goodness of fit for logistic mode – the c-statistics ( or area under the ROC) is preferable.

7. PLOS authors have the option to publish the peer review history of their article (what does this mean?). If published, this will include your full peer review and any attached files.

Reviewer #2: Yes: Manop Kanato, Ph.D. Associate Professor, Department of Community Medicine, Khon Kaen University, Thailand. President of Administrative Committee of Substance abuse Academic Network, Office of the Narcotic Control Board of Thailand.

Reviewer #4: No

---

## [Author Response · Author response to Decision Letter 1]

24 May 2020

Reviewer #2: Although the author respond to comments, it is unclear.

1.Sampling method was straified cluster sampling, thus sample size calculation needs adjustment for design effect.

: Thank you very much professor for such great comment. We have tried to calculate the sample size by putiing the effect size, however, it is still in around 700. Moreover, the from the CIs particularly in presentation the significant, there were presented with very narrow such as in sex; OR=4.75, 95%CI=2.27-9.95, and aged 21-24 years; OR=2.51, 95%CI=1.11-5.71. 

: Moreover, this is a cross-sectional without any previous information particularly in the prevalence of MA use in this population, therefore, we have carefully discussed with two statisticians; one from Mahidol University and another one is my professor from Harvard School of Public Health, both them suggested that with the assumptions, it is good to use the samples obtained in the study. 

: However, we will keep in mind this significant point from your comments for our next work. Thank you very much professor.

2.Reliability process is unusal (three rounds with the same 20 samples and calulated alpha from last round). Recall bias appeared. Alpha should be presented for individual section rather than overall.

: This is common while developing the quality of the tool. We can not calculate the the reliability in the first round because after finishing the first round, many points were changed and improved. It meat that we need to get the final and completed version to test and calculate for reliability score before use. 

: Another very important point is this is the pioneer project on doing in the MA problem among the hill tribe who have limited in use Thai. Then, we had had very carefully obtained the information and also approached tham.

: I very hope that you understand us and thank you very much.

3. Table 2-5 should be compare between two hill-tribes.

: thank you for the comment, it’s revised and improved.

4. Although the MA lifetime prevalence is 3 times higher than general population of Thailand in 2019, it should be dicussed why it is. The specific culture of these hill-tribes affected to MA use still need to clarify.

: Thank you very much for the comment. It’s improved in page no.

Reviewer #4: The authors were responsive to comments and the paper is now improved and has greater clarity overall. However, not all the relevant responses that are given in the letter to the editor are included in the paper. I have a few remaining comments:

Specific comments:

1. In the abstract, conclusions: the last sentence and intervention for “male youth” need to be better explained (I suppose as a target population) – this sentence is still not well phrased. is the intention to say: …interventions that lowers risk of MA use by addressing family relationship, male youth risk behaviors….etc….?

: Thank you for comment, it’s improved.

2. Sample size calculation: still need clearer description of what was done under what assumption. if the calculations were to estimate the proportion of MA use, then a single sample with confidence interval widths could be used. if it is based on comparing two proportions (which is what is indicated), then, explain which groups are compared and if this is within each tribe group? - this may not be a critical point in the paper but if included, it should be clear and relevant to the analysis performed.

: Thank you so much for the valuable comment in this point. It has been revised an improved in section of sample size calculation.

: Since, there is no scientific data available on the prevalence of the MA use among the hill tribe population, then, the calculation for the sample size was based on the information (prevalence) from the study conducted in Thai youth who lived in the central of Bangkok which was conducted by Toeam, et al [26]. Moreover, based on the information of the number of population between the Akha and Lahu which was reported by the Hill tribe Welfare and Development Center [18], two tribes had similar size of the population living 243 Akah villages (approximately 60,000 population) and 216 Lahu villages (approximately 50,000 population). 

3. As more than one youth per household could participate, can the authors add the info on how many clusters with size>1 were included? as the analysis does not take this clustering into account at a minimum, a note referring to potential biases in estimating the statistical significance (standard errors of estimates) should be included in methods and/or discussion. this is an important methodological issue.

: We have revised our raw data, and it was found that no household or family that presented more than one participant. We accept that we never though this issue before. We have learned this new point, thank you very much.

: However, we have put information in section of “step of data collection” to make clear the point.

4. Results: I do not find the reported “response rate” to the survey in the manuscript.

: We have responded to this point in our previous version regarding to the comment from one of reviewers. Please see the first sentences on the result section.

5. Table 1-5: the paper would benefit by creating tables 1-5 that contain more information including: a) the original n & 5 columns for overall frequencies and combining with it the univariate analysis that appears in table 6. then table 6 would just have the final multivariate table. this will be make it easier for the reader to capture the final model.

:Thank you very much for the comment. However, the previous reviewer suggested to put more statistics in table no.2-5 (Comment no.2). Therefore, we would like to maintain on univariate and multi variate analyses in same table (table 6). It’s also much more easier in explain the relationship between variables. I very hope that you understand us. 

6. Description on how the MV model developed should be added to methods (it is in the response letter but not in the paper)

: Thank you very much for the comment. It’s improved in page no. 6

7. for goodness of fit for logistic mode – the c-statistics (or area under the ROC) is preferable.

: Thank you very much. This is one thing that I have learned thank you so much.

Thank you so much!

TK

Regards,

TK

Assistant Professor Dr.Tawatchai Apidechkul

Deputy Dean, School of Health Science, MFU

Director, Center of Excellence of the Hill tribe Health Research, WHO-CC 

Former Hubert H Humphrey Fellow (2013-2014), Emory University

Global Health Delivery Intensive (Harvard School of Public Health)

---

## [Editor Report · Decision Letter 2]

2 Jun 2020

PONE-D-20-01893R2

Associations of childhood experiences and methamphetamine use among Akha and Lahu hill tribe youths in northern Thailand: A cross-sectional study

PLOS ONE

Dear Dr. Apidechkul,

Thank you for submitting your manuscript to PLOS ONE. After careful consideration, we feel that it has merit but does not fully meet PLOS ONE’s publication criteria as it currently stands. Therefore, we invite you to submit a revised version of the manuscript that addresses the points raised during the review process.

We look forward to receiving your revised manuscript.

Kind regards,

Siyan Yi, MD, MHSc, PhD

Academic Editor

PLOS ONE

Additional Editor Comments (if provided):

Thanks for your revisions. The manuscript is now much improved and almost ready for publication. However, since PLOS ONE does not allow you to proofread your manuscript after acceptance, I would like you to take this opportunity to do so. Here are some examples for your consideration:

1. I am not sure if ‘Thai-Myanmar-Laos Republic’ border is correct. If the ‘Republic’ is for Laos, the more commonly used is ‘Lao People's Democratic Republic’ or just ‘Laos.’

2. You may consider using ‘adverse childhood experiences’ which is widely used in the literature instead of ‘bad childhood experiences.’

3. Abstract:

- First sentence in Methods may be revised to avoid repeating the objective.

- Results: After controlling for…

- Since the analyses included both Akha and Lahu youth, repeating the expression ‘…among Akha and Lahu youth in northern Thailand’ brings more confusions than helps and unnecessarily increased the word count.

- I am not sure if ‘ORadj’is commonly used. May consider ‘adjusted odds ratio (AOR).

4. Methods:

- It would great if you could add a little bit more information on the inclusion criteria for both youth (participants) and parents. More information of the selection of the participants (youth and parents) is also required.

- Since this study was conducted among tribal populations, research instrument should also include information on the languages used for the questionnaire for each tribe, if translation (and back-translation) was performed, and average time required for the interview.

- ‘The questionnaires were conducted three (3) times in the same piloting samples…’ This is hard to understand: what does this mean? How many questionnaires were developed? If different questionnaires were developed for youth and parents, this should be clearly described.

- The flow of the information would be better if the sampling procedures (Steps of data collection) comes before ‘Research instruments.’

- ‘All questionnaires were properly destroyed after coding’ – what does this really mean? After data entry?

- The writing of ‘Statistical analysis’ can be improved by removing typos and grammatical errors. ‘Conceptual framework’ was mentioned without earlier discussion.

5. Please proofread the whole manuscript accordingly.

---

## [Author Response · Author response to Decision Letter 2]

3 Jun 2020

Additional Editor Comments (if provided):

Thanks for your revisions. The manuscript is now much improved and almost ready for publication. However, since PLOS ONE does not allow you to proofread your manuscript after acceptance, I would like you to take this opportunity to do so. Here are some examples for your consideration:

: Thank you so much!

1. I am not sure if ‘Thai-Myanmar-Laos Republic’ border is correct. If the ‘Republic’ is for Laos, the more commonly used is ‘Lao People's Democratic Republic’ or just ‘Laos.’

: Thank you so much for the correction in the point, it’s replaced by “Laos” in whole tex.

2. You may consider using ‘adverse childhood experiences’ which is widely used in the literature instead of ‘bad childhood experiences.’

: As we have responded to the point in previous version that in this study that the cross-sectional was used to explored on all factors relevant to MA use in late years of the Akha and Lahu youths. Having bad experience while very early ages were determined as one of factors in the model. Therefore, we have decided that if it is used the word of “adverse childhood experience” may not reflect the study concept. 

: However, we concern a lot and have had long discussion in the point. 

: Thank you very much for your concern, and now we are planning a new project to detect a particular of impact of “adverse childhood experience” among these population by a stronger study design.

: Thank you once again for your valuable suggestion.

3. Abstract:

- First sentence in Methods may be revised to avoid repeating the objective.

: Thank you so much, it’s improved.

- Results: After controlling for…

: Thank you, it’s improved

- Since the analyses included both Akha and Lahu youth, repeating the expression ‘…among Akha and Lahu youth in northern Thailand’ brings more confusions than helps and unnecessarily increased the word count.

: Thank you, we agree with you and it’s improved.

- I am not sure if ‘ORadj’is commonly used. May consider ‘adjusted odds ratio (AOR).

: Replaces all

4. Methods:

- It would great if you could add a little bit more information on the inclusion criteria for both youth (participants) and parents. More information of the selection of the participants (youth and parents) is also required.

: It’s improved. Detail of selection the participants is provided in the “Step of data collection”

- Since this study was conducted among tribal populations, research instrument should also include information on the languages used for the questionnaire for each tribe, if translation (and back-translation) was performed, and average time required for the interview.

: Thank you for great comment, it‘s improved. In the section of language, the questionnaire is provided in Thai because all of the participants including their parents are able to use Thai. 

- ‘The questionnaires were conducted three (3) times in the same piloting samples…’ This is hard to understand: what does this mean? How many questionnaires were developed? If different questionnaires were developed for youth and parents, this should be clearly described.

: Since this is the pioneer of the project relevant to the MA use in the hill tribe, therefore, no standard or other questionnaire are available. 

In this step (pilot), we had several purposes to do this such as sequencing of the questions, words or sentences used weather agiant culture or belief, or feel stigma or not, questions make free of sense to answer or not. Therefore, it’s needed to have many times to repeat for making sure the quality of the questionnaire is met. 

: The questions for children, were tested among the youths and the questions for the parents were asked their parents accordingly.

: We developed only one set of questionnaires, with having six parts, presented detail in page. 4-5

: We very hope you understand our situation.

- The flow of the information would be better if the sampling procedures (Steps of data collection) comes before ‘Research instruments.’

: Thank you, it’s re-sequence accordinly

- ‘All questionnaires were properly destroyed after coding’ – what does this really mean? After data entry?

: It means that all filled questionnaire form (hard copy by participants and parents, there were destroyed after coding and putting into the computer by cutting it into a very small piece and burning with the university waste management processing. This reflects on the comments of the Ethical Consideration Board that to protect the release of any information to public which we has strictly followed the comments. 

- The writing of ‘Statistical analysis’ can be improved by removing typos and grammatical errors. ‘Conceptual framework’ was mentioned without earlier discussion.

: Thank you so much, it’s improved.

5. Please proofread the whole manuscript accordingly.

: We all (authors) have carefully looked throughout the whole paper at least three rounds and many points have been improved. Thank you so much.

I have uploaded two files as supplements; questionnaire and data file of the study. 

Thank you so much!

TK

---

## [Editor Report · Decision Letter 3]

5 Jun 2020

Associations of childhood experiences and methamphetamine use among Akha and Lahu hill tribe youths in northern Thailand: A cross-sectional study

PONE-D-20-01893R3

Dear Dr. Apidechkul,

We’re pleased to inform you that your manuscript has been judged scientifically suitable for publication and will be formally accepted for publication once it meets all outstanding technical requirements.

Kind regards,

Siyan Yi, MD, MHSc, PhD

Academic Editor

PLOS ONE
---

## [Editor Report · Acceptance letter]

9 Jun 2020

PONE-D-20-01893R3 

Associations of childhood experiences and methamphetamine use among Akha and Lahu hill tribe youths in northern Thailand: A cross-sectional study 

Dear Dr. Apidechkul:

I'm pleased to inform you that your manuscript has been deemed suitable for publication in PLOS ONE. Congratulations! Your manuscript is now with our production department. 

Kind regards, 

on behalf of

Dr. Siyan Yi 

Academic Editor

PLOS ONE